# Seven centuries of reconstructed Brahmaputra River discharge demonstrate underestimated high discharge and flood hazard frequency

Mukund P. Rao [1,2✉], Edward R. Cook[1], Benjamin I. Cook[3,4], Rosanne D. D'Arrigo[1], Jonathan G. Palmer [5], Upmanu Lall[6], Connie A. Woodhouse [7], Brendan M. Buckley[1], Maria Uriarte [8], Daniel A. Bishop [1,2], Jun Jian [9] & Peter J. Webster[10]

The lower Brahmaputra River in Bangladesh and Northeast India often floods during the monsoon season, with catastrophic consequences for people throughout the region. While most climate models predict an intensified monsoon and increase in flood risk with warming, robust baseline estimates of natural climate variability in the basin are limited by the short observational record. Here we use a new seven-century (1309–2004 C.E) tree-ring reconstruction of monsoon season Brahmaputra discharge to demonstrate that the early instrumental period (1956–1986 C.E.) ranks amongst the driest of the past seven centuries (13th percentile). Further, flood hazard inferred from the recurrence frequency of high discharge years is severely underestimated by 24–38% in the instrumental record compared to previous centuries and climate model projections. A focus on only recent observations will therefore be insufficient to accurately characterise flood hazard risk in the region, both in the context of natural variability and climate change.

[1] Tree Ring Laboratory, Lamont-Doherty Earth Observatory of Columbia University, Palisades, NY 10964, USA. [2] Department of Earth and Environmental Science, Columbia University, New York, NY 10027, USA. [3] NASA Goddard Institute for Space Studies, New York, NY 10025, USA. [4] Ocean & Climate Physics, Lamont-Doherty Earth Observatory of Columbia University, Palisades, NY 10964, USA. [5] ARC Centre of Excellence in Australian Biodiversity and Heritage, School of Biological, Earth and Environmental Sciences, University of New South Wales, Sydney, NSW 2052, Australia. [6] Department of Earth and Environmental Engineering, Columbia University, New York, NY 10027, USA. [7] School of Geography and Development, University of Arizona, Tucson, AZ 85721, USA. [8] Ecology, Evolution, and Environmental Biology, Columbia University, New York, NY 10027, USA. [9] Dalian Maritime University, Dalian 116024, China. [10] Earth and Atmospheric Sciences, Georgia Institute of Technology, Atlanta, GA 30318, USA. ✉email: mukund24rao@gmail.com

The Brahmaputra River contributes nearly half of the ~40,000 m³/s mean annual discharge of the Ganga–Brahmaputra–Meghna river system (Fig. 1). This makes it the joint third largest river system in the world (tied with the Río Orinoco, Venezuela) in terms of its mean annual discharge after the Amazon and Congo[1]. Known as the Jamuna in Bangladesh, the high discharge rates of the Brahmaputra are caused, in part, by annual precipitation (rain and seasonal snow) in excess of 3000 mm/year for much of the watershed (Supplementary Fig. 1) and snowmelt from its highly glaciated upper basin encompassing the Eastern Himalaya and parts of the Southern Tibetan Plateau[2–6]. The river and its tributaries provide important societal, ecological, cultural, and economic services to more than 60 million people in Bangladesh, North-eastern India, Bhutan, and Tibet, China[1,7]. These benefits include fish (a primary source of protein in the region), water to irrigate many seasonal rice varieties that need annual flood waters to survive, the deposition of fresh sediment to sustain the large inhabited riverine islands (known as chars), and the prevention of salt-water intrusion from the Bay of Bengal into the low-lying Sundarban delta[7–9].

Although the Brahmaputra River provides these important benefits, it is also a frequent cause of human suffering from flooding in Bangladesh and Northeast India (primarily in Assam)[10,11]. Long-duration (>10-day) floods that cause the most widespread disruptions are most common during the monsoon season between July and September[11–14]. The main driver of monsoon season July–August–September (JAS) discharge in the Brahmaputra is upper basin precipitation (Fig. 1a and Supplementary Figs. 2–4), along with smaller contributions from glacial melt, snow melt, and base flow[15,16]. For example, the year 1998 witnessed intense monsoon flooding between July and September in both Bangladesh and Assam, inundating nearly 70% of Bangladesh, affecting over 30 million people and causing a humanitarian emergency in the region[12,13,15,17]. Similar floods in 1987, 1988, 2007, and 2010 along with the currently ongoing inundation from flooding in 2020 have

caused large fatalities, permanent loss of livelihoods, and the displacement of thousands of people to urban centres like Dhaka, in addition to raising regional food security concerns due to famine from damaged crops[12,13,18,19].

While anthropogenic sulphate aerosol emissions caused a reduction in South Asian Summer Monsoon (SASM) activity during the latter half of the twentieth century[20–23], increasing carbon-dioxide emissions and decreased aerosol loading are projected to intensify the South Asian Summer Monsoon through the twenty-first century[24]. This intensification of the monsoon, along with the accelerated warming-driven glacial melt, is expected to lead to greater flow in the Brahmaputra River[3,16] and likelihood of flood hazard in the region[14,25–27].

Studies of long-term flood hazard in the Brahmaputra watershed have, however, been hampered by the relatively short and fragmentary instrumental discharge records available[15,28]. The longest instrumental record of Brahmaputra discharge comes from the Bahadurabad gauging station in Bangladesh, shortly after the river enters the country from Assam, India (Fig. 1). The Bahadurabad discharge record spans about six decades from 1956 to 2011, interspersed with some missing data (Figs. 2 and 3a). Such a short record makes it difficult to assess and put into perspective the magnitude of projected future changes relative to natural variability, especially at decadal and centennial timescales[15,24].

Tree-ring reconstructions of hydroclimate (including streamflow) are used to extend instrumental records to evaluate the severity of past droughts and pluvials[29–34], as a reference to interpret recent climate extremes relative to those in the past[35–37], and to contextualise natural climate variability in the system[38] relative to climate change projections[35,39,40]. To that end, we develop a monsoon season reconstruction of mean JAS Brahmaputra River discharge at Bahadurabad, Bangladesh.

We use our reconstruction, along with historical documentation of flood events[10,11] to evaluate the connections between discharge and monsoon season flooding. We then derive

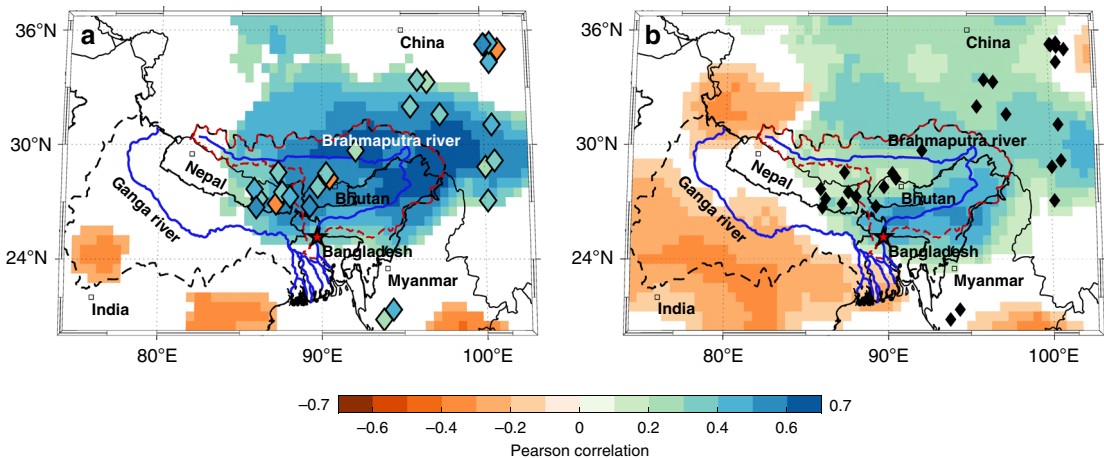

**Fig. 1 Climate-streamflow-tree-growth relationships in the Brahmaputra watershed (red dashed lines).** The figures also highlight the larger Ganga–Brahmaputra–Meghna watershed (black dashed line), and the locations of the 28 tree-ring predictors (diamonds) used in the mean July–August–September (JAS) streamflow reconstruction at the Bahadurabad gauge, Bangladesh (red star). **a** Infill shading in diamond markers represent the Pearson correlation between mean JAS discharge at Bahadurabad and each tree ring predictor (1956–1998 C.E.). Background shading is the spatial field correlation between mean JAS discharge at Bahadurabad and mean JAS precipitation (1956–2011 C.E.) **b** Spatial field correlation between the first principal component (PC1) of the 28 tree ring predictors (variance explained: 24.86%) and mean JAS precipitation (1956–1998 C.E.). Spatial correlations in (**a**) and (**b**) are against CRU Ts 4.01 precipitation. Together, (**a**) and (**b**) show that monsoon season JAS flow in the Brahmaputra is positively related to upper basin precipitation in a region largely co-located with the tree ring predictor network. They also demonstrate that the predictor network effectively captures regional JAS precipitation independent to its correlation with JAS Brahmaputra discharge. Note that the locations of predictors are jittered for display. Only correlations significant at $p < 0.05$ using a 2-tailed $t$-test are shown. See Supplementary Table 1 for more information on the predictor network and Supplementary Fig. 2 for similar analyses with GPCC precipitation.

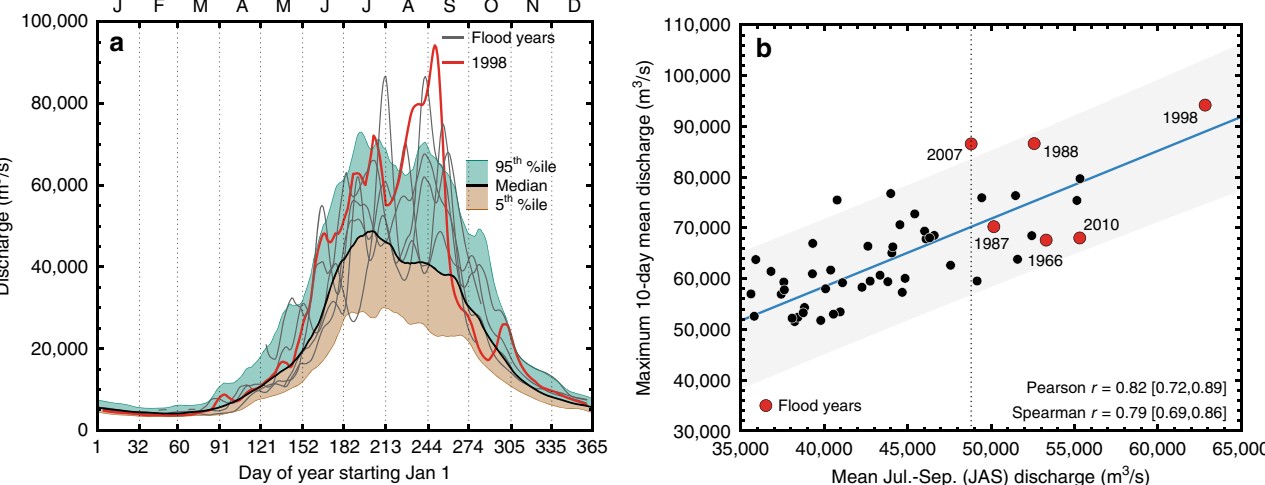

**Fig. 2 Discharge characteristics of the Brahmaputra at Bahadurabad, Bangladesh between 1956 and 2011 C.E. a** Annual 10-day mean discharge hydrograph. The brown and green envelopes represent the 5th, 50th, and 95th percentiles of 10-day mean discharge, respectively. The 5 grey and 1 red line represent 10-day mean discharge during instrumental period flood years in 1966, 1987, 1988, 1998 (in red), 2007, and 2010 C.E. **b** Scatter plot of mean JAS discharge against maximum 10-day mean daily discharge. The six flood years are highlighted in red. The vertical dashed line highlights mean JAS discharge in 2007 (48,800 m³/s), the lowest discharge of the 6 instrumental period documented flood years. The bootstrapped Pearson and Spearman rank correlations are calculated as the median and 5th and 95th percentile of 1000 draws with replacement. The grey uncertainty envelope (±2σ) is derived from the best-fit linear regression (blue line).

projections of future Brahmaputra River discharge from climate model simulations (historical and RCP8.5) participating in the fifth phase of the Coupled Model Intercomparison Project (CMIP5—ref. [41]). We use these models and our reconstruction to evaluate two situations. The first is how the recurrence of high discharge events (used here as a proxy for flood hazard) in recent decades compares to longer-term estimates over the last several centuries. The second is how the increase in Brahmaputra River discharge caused by projected regional wetting[16] compares to natural climate variability estimated by the instrumental data and our tree-ring derived reconstruction. In this paper we define flood hazard as an exceedance of mean JAS discharge of 48,800 m³/s corresponding to observed discharge in 2007, the lowest discharge of 6 instrumental period flood years between 1956 and 2011.

## Results

**Seasonal hydrograph and recent flood events in the observed record.** The most severe monsoon-season floods by the Brahmaputra River are those that cause inundation for more than 10 consecutive days. Such floods most commonly occur during JAS, the season with largest discharge during the year[13]. The annual maximum 10-day mean discharge hydrograph at Bahadurabad between 1956 and 2011 shown in Fig. 2a describes the evolution of discharge from a dry, low-flow period between November of the prior year through May of the current year, and a period of peak discharge during the monsoon season in JAS[42]. As tree rings typically provide information regarding seasonal hydroclimate[43], we then attempted to determine whether JAS monsoon season discharge is related to sub-seasonal flow at the 10-day timescale relevant to regional flooding. We found a strong and significant positive relationship between mean JAS seasonal discharge and the maximum 10-day mean discharge in each year (Spearman $r = 0.79$; Pearson $r = 0.82$; $n = 55$, $p < 0.001$, Fig. 2b). This coupling between 'instantaneous' and 'seasonal' discharge is a common feature among large river basins across the world, including the Brahmaputra[28,44].

This relationship is further supported by comparing the 10-day mean discharge hydrograph of six instrumental period flood years (i.e. 1966, 1987, 1988, 1998, 2007, and 2010) to the overall 10-day

mean discharge hydrograph between 1956 and 2011 (Fig. 2a, also see ref. [15]). In each of these known flood years, 10-day mean discharge exceeded the 50th and 95th percentile of daily discharge for an extended duration. The median number of discharge days that exceeded the 50th percentile and 95th percentile of daily JAS discharge during these flood years were 76 and 17, respectively, representing 82% and 18% of the total of 92 days in the 3-month JAS period (Supplementary Fig. 5). In particular, in the year 1998, peak 10-day mean discharge was ~94,000 m³/s. This was greater than twice the median flow for the season[1,12,15], with daily flow exceeding the 95th percentile of daily flow for a total of 39 of the 92 days, or 42% of the days in the JAS season. These results confirm the use of JAS discharge as a proxy for 10-day discharge and as an indicator for the likelihood of flood hazard.

**Predictor selection and reconstruction model fidelity.** To extend the short instrumental record, we developed a reconstruction of mean JAS monsoon season discharge extending from 1309 to 2004 C.E. (Common Era) for the Brahmaputra River at Bahadurabad, Bangladesh, using Bayesian Regression (see 'Methods' section). We used a pool of 28 annually dated tree-ring series that were proximally located to the Brahmaputra watershed boundary (Supplementary Table 1). All chosen series correlate well with mean JAS streamflow at Bahadurabad ($p < 0.10$, using a 2-tailed $t$-test) during the 1956–1998 model calibration-validation period. Of these 28 predictors, 5 are located relatively distant (~670 km) to the watershed. We retained these 5 predictors as they contributed additional model skill, as is commonly found in tree-ring hydroclimate and streamflow reconstructions[31,45–50], consistent with spatial autocorrelation in climate fields[51–53].

The first principal component (PC1) of the 28 tree-ring predictors explained 24.86% of the total variance in the set of predictor chronologies. The PC1 timeseries of these predictors also correlated significantly with JAS discharge at Bahadurabad between 1956 and 1998 ($r = 0.73$, $n = 42$, $p < 0.01$, 2-sided $t$-test), and with mean JAS precipitation in the Brahmaputra basin (Fig. 1b and Supplementary Fig. 2b). The loadings of each predictor on PC1 (and PC2) are described in Supplementary Table 1. The correlation between the PC1 series and regional

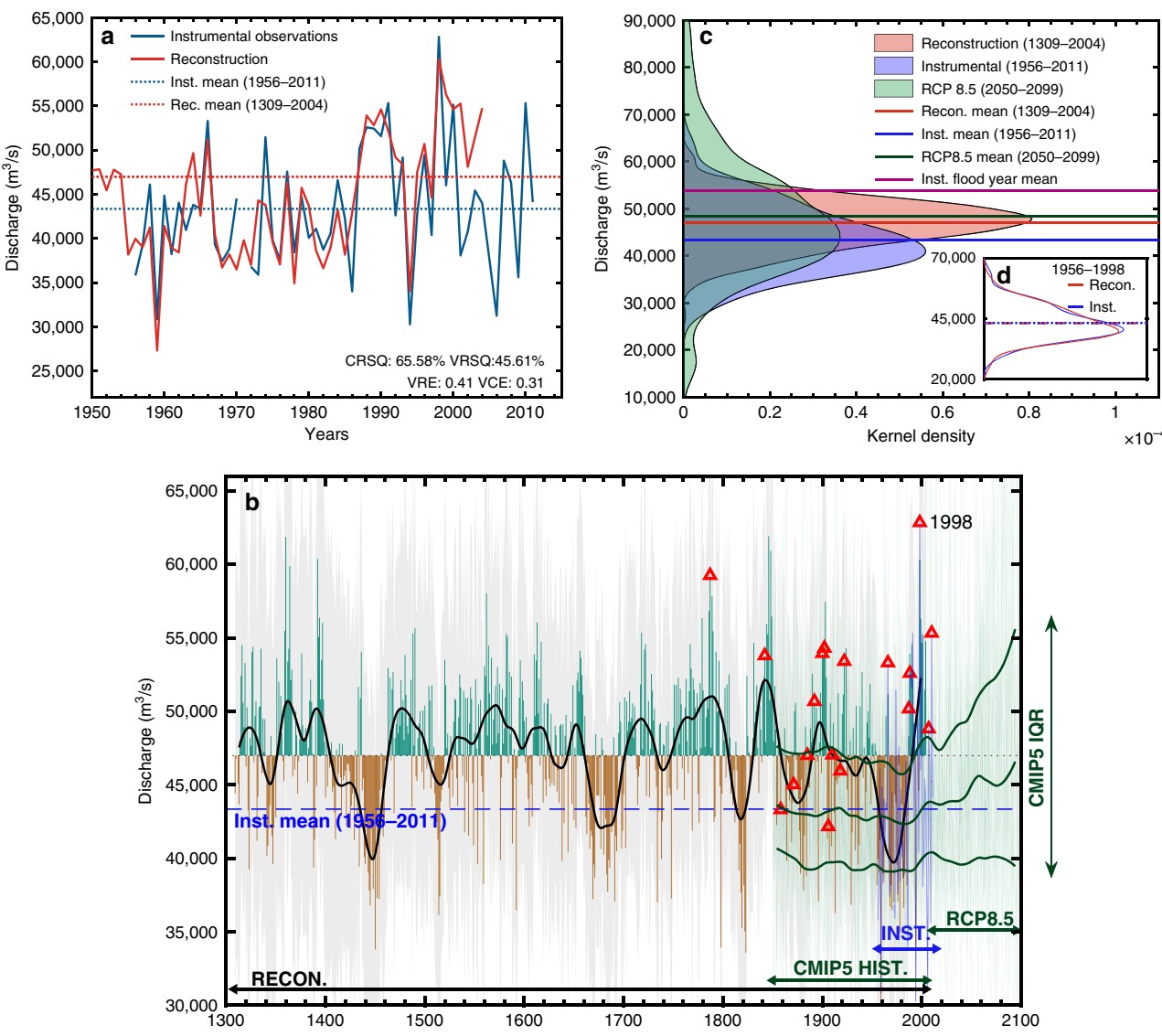

**Fig. 3 Instrumental observations, the reconstruction, and CMIP5 projections of mean JAS discharge at Bahadurabad. a** JAS instrumental discharge and its mean (43,350 m³/s) compared against reconstructed JAS discharge and its long-term mean (46,993 ± 812 m³/s). **b** Reconstructed discharge for each year between 1309 and 2004 C.E. as a departure from the reconstructed mean (as green and brown bars), along with the 50-year low-pass filtered reconstruction (solid black) highlighting multi-decadal variability. The instrumental JAS discharge and its mean between 1956 and 2011 C.E. is shown in the blue and dashed blue lines, respectively. Red triangles mark 18 documented flood years between 1787 and 2010 C.E. The 3 dark green lines represent the 50-year low-pass filtered interquartile range (IQR—25th, 50th, and 75th percentiles) of the multi-model CMIP5 RCP8.5 ensemble (20 models; 42 runs; Supplementary Table 2) along with the full range of variability (light green lines) during both the 'historical' (1850–2005 C.E.) and 'future' (2006–2099 C. E.) simulation period of these runs. **c** Kernel density profiles of the median reconstruction (in red), instrumental period (in blue), the full 42 member CMIP5 RCP8.5 end of the century simulation period (2050–2099 C.E.) ensemble suite (in green) and their respective means. The observed mean discharge of the 6 instrumental period flood years from Fig. 2 are shown in purple. The inset figure **d** shows the kernel density profiles of mean JAS instrumental discharge (in red) and reconstructed mean JAS discharge (in blue) over the calibration-validation period (1956–1998 C.E.) along with their means. The reconstruction matches the features of instrumental discharge such at its mean and variance in this period.

precipitation in Fig. 1b indicates that the 'shared variance' between the selected predictors captures regional hydroclimate variability independent to their statistical relationship with Brahmaputra streamflow.

Median calibration-validation statistics of our reconstruction between 1309 and 2004 C.E. are as follows: i. CRSQ (calibration period coefficient of multiple determination): 65.58%, ii. VRSQ (validation period square of the Pearson correlation): 45.61%, iii. VRE (validation period reduction of error): 0.41, and iv. VCE (validation period coefficient of efficiency): 0.31. A comparison of the correlations between reconstructed discharge and upper basin

climate variables indicates that the reconstruction captures the climate-streamflow relationships inherent in the instrumental observations (Supplementary Fig. 4). However, these climate-streamflow relationships are slightly weaker for reconstructed discharge than for instrumental discharge. The calibration-validation statistics for each model nest are shown in Supplementary Fig. 6 along with the number of tree ring predictors used in each nest (maximum 28, minimum 10). That VRE and VCE values are consistently greater than zero across the full reconstruction period indicates that the model has some skill[32].

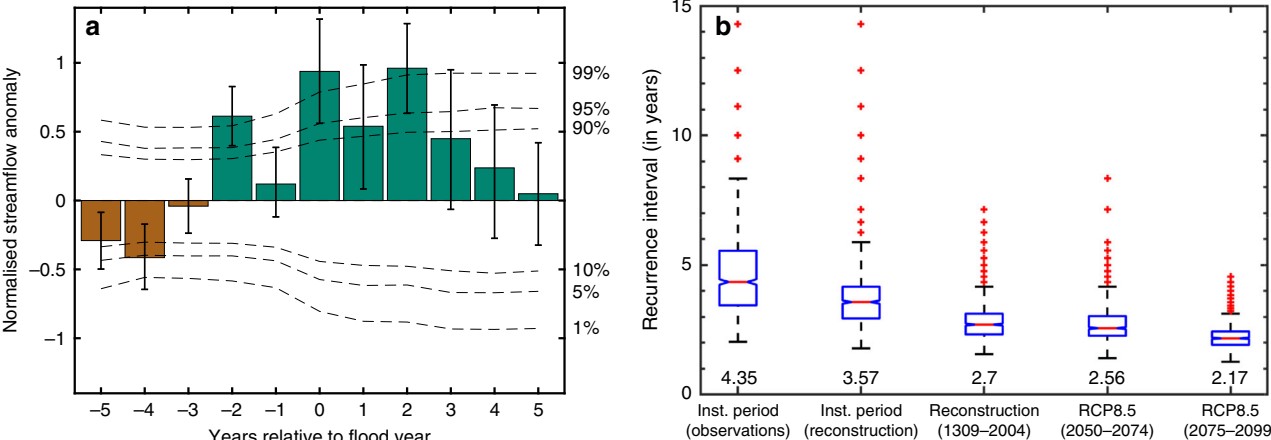

**Fig. 4 Discharge characteristics of wet and dry periods and flood years. a** Superposed epoch analysis (SEA) showing higher than normal flows in historical and instrumental period flood years (in year t+0) between 1780 and 2004 C.E. than would be expected by chance. The response is the 5th, 50th, and 95th percentiles of mean flow across 1000 unique draws of 10 flood years at random out of 16. The horizontal dotted lines indicate the threshold required for epochal anomalies to be statistically significant using random bootstrapping at three different statistical thresholds. These thresholds were calculated by compositing 10,000 draws of 10 years at random (or 'pseudo-flood years') from the reconstruction between 1780 and 2004 C.E. **b** Recurrence intervals (in years) of discharge greater than the 2007 C.E. flood year JAS discharge calculated from 1000 draws of 30 years with replacement form the instrumental data (1956–2011 C.E.), the reconstructions (over the instrumental period, 1956–2004 C.E.), and the full 42 ensemble member CMIP5 RCP8.5 simulation suite between 2050–2074 C.E. and 2075–2099 C.E. The median recurrence interval for each dataset is noted below each boxplot. The median recurrence interval for instrumental discharge between 1956–2004 C.E. and 1956–1998 C.E. remains 4.35 (Supplementary Fig. 11).

**Reconstruction of past discharge**. The reconstructed discharge estimates during the observational period are shown in Fig. 3a and our full reconstruction between 1309 and 2004 C.E. is presented in Fig. 3b. While the reconstruction is calibrated to the instrumental mean and variance in the reconstruction procedure (Fig. 3c, d), we found that the mean reconstructed discharge over the full reconstructed period between 1309 and 2004 C.E. was significantly higher than the instrumental mean between 1956 and 2011 C.E. ($46,993 \pm 812$ m$^3$/s cf. $43,350$ m$^3$/s, difference of means = $3,644$ m$^3$/s, $t$-statistic = $5.11$, $p < 0.01$) (Fig. 3a). The uncertainty range around the mean was derived from the 5th and 95th percentiles of the means across all 400 iterations of the median reconstruction. The reconstruction was also significantly ($p < 0.01$) wetter than the instrumental period even if we used the 1956–2004 C.E. instrumental mean of $43,442$ m$^3$/s or the 1956–1998 C.E. calibration-validation period instrumental mean of $43,233$ m$^3$/s. This difference between the instrumental and reconstructed mean is largely driven by the first three decades of the instrumental observations between 1956 and 1986 C.E. that was unusually dry in the long-term context of the past seven centuries[54]. The instrumental mean between 1956 and 1986 C.E. was $41,206$ m$^3$/s. We also compared mean JAS discharge during these 4 intervals (i.e. 1956–1986, 1956–1998, 1956–2004, and 1956–2011) to random block bootstrap draws from the full reconstruction. We found all four intervals to be significantly drier ($p < 0.05$) than the reconstruction (Supplementary Fig. 7).

The 31-year 1956–1986 C.E. instrumental mean of $41,206$ m$^3$/s ranked in the 13th percentile of our 696-year reconstruction. The 1956–2004 C.E. mean of $43,233$ m$^3$/s and the full 56-year 1956–2011 C.E. instrumental mean of $43,350$ fell in the 22nd percentile of the full reconstruction. These results highlight the unusually dry nature of the modern instrumental period relative to the full reconstructed record, although we do observe dry periods of similar or greater magnitude in the early 1400s, late 1600s, early 1800s, and late 1800s (also see Supplementary Fig. 8, refs. [54,55] for perspectives on upper Brahmaputra watershed May-June hydroclimate, and refs. [56,57]). On the other hand, the reconstruction indicates long multidecadal wet periods of above normal discharge between ~1560–1600 C.E., 1750–1800 C.E., and

~1830–1860 C.E. that have no analogues in the instrumental data (Fig. 3 and Supplementary Fig. 8, also see ref. [55]). We do note that both instrumental observations and the reconstructions show a return to wetter conditions starting in 1987 (also see refs. [58,59]). For example, more recent instrumental observations of discharge between 1987–1998 C.E., 1987–2004 C.E., and 1987–2011 C.E. are relatively wetter than the instrumental data prior to 1987 and fall in the 59th, 48th, and 39th percentile of full reconstruction (Supplementary Fig. 7). However, the lack of more up-to-date streamflow data preclude us from being able to contextualise more recent years of discharge relative to the longer term reconstructed mean.

**Historical flood events**. Next, we evaluated the relationship between discharge and 12 historical Brahmaputra flood years in 1787, 1842, 1858, 1871, 1885, 1892, 1900, 1902, 1906, 1910, 1918, 1922 C.E. identified from refs. [10–12] and 6 recent instrumental period flood years in 1966, 1988, 1987, 1998, 2007, 2010 C.E.[15]. These flood years (18 total) are marked as red triangles in Fig. 3b. The mean observed JAS discharge during these six instrumental period flood years was ~$53,800$ m$^3$/s (purple line in Fig. 3c). The reconstructed discharge of $60,312$ m$^3$/s in 1998 C.E. that was concurrent with large scale flooding in Bangladesh is only exceeded five times in the 696-year reconstruction. While our reconstruction model underpredicts the instrumental discharge of $62,840$ m$^3$/s in 1998 C.E. by ~$2500$ m$^3$/s (Fig. 3a), this comparison still suggests that within the reconstruction, discharge that year was unusually high[12,15] even in the long-term context of the past seven centuries.

Following this, we tested the probability of random association between the 16 flood years and high discharge over the reconstruction period (excluding 2007 and 2010 as the reconstruction ends in 2004) using superposed epoch analysis (SEA—ref. [60]; see 'Methods'). We found that mean reconstructed flows across these 16 years are significantly wetter ($p < 0.001$) and approximately one standard deviation higher than would be expected by chance (Fig. 4a). This result is consistent with our finding using the instrumental discharge data (Fig. 2) that flood

events co-occur with high seasonal JAS discharge. However, we found this relationship to be much weaker in the reconstructions. Six of the 12 pre-1956 C.E. flood events occurred in relatively dry years. SEA of the 12 flood years prior to 1956 did show a median result of wet conditions in the year in which a flood was documented (Supplementary Fig. 9). However, this result was not statistically significant at $p < 0.05$ despite the 95th percentile of bootstrapped discharge responses being significant at $p < 0.001$. This asymmetric response may be partly due to some of these historical flood years being undivided Bengal (Bangladesh, and West Bengal, India) flood years and not solely Brahmaputra flood years, and lack of information regarding the magnitude of flooding in the historical sources that we used.

**Projections of future discharge**. We calculated projections of runoff for the Brahmaputra River at Bahadurabad using an ensemble of 20 CMIP5 climate models (42 ensemble members) that provided continuous simulations from 1850 through 2099 C. E. (historical simulation from 1850–2005 C.E.; high emissions RCP8.5 scenario from 2006–2099 C.E.) (Supplementary Table 2). During the historical simulation period, the multi-model ensemble interquartile range (IQR—25th, 50th, and 75th percentiles) of the 20 climate models shows a decreasing trend from ~1940–1980 C.E. (shown in solid green, Fig. 3b), with a recovery in discharge between 1980–2005 C.E. Future projections of the IQR of multi-model ensemble discharge suggest a large increase in discharge relative to the instrumental mean starting ~2025 C.E. that is expected to persist and intensify through to the end of the century.

We find that towards the end of the century, between 2050 and 2099 C.E., the 25th percentile of CMIP5 multi-model discharge remains relatively constant but there are large increases in the 50th and 75th percentiles of projected discharge. This can also be observed in a comparison of the kernel density profile of all 42 ensemble members of CMIP5 RCP8.5 scenario discharge projections between 2050 and 2099 C.E., compared to the kernel density profiles of both the instrumental data between 1956 and 2011 C.E. and the full period of the reconstruction between 1309 and 2004 C.E. (Fig. 3c). The kernel density profile of discharge over instrumental observations between 1956 and 2011 C.E. and the horizontal line representing the instrumental mean in this period also illustrate that the instrumental observations are drier than the long-term mean variability in the river system suggested by the reconstruction, and likely drier than future projected runoff.

**High discharge related flood hazard relative to instrumental observations**. We then calculated the difference in the likelihood of high discharge in the instrumental observations (1956–2011 C. E.) against the final median reconstruction (1309–2004 C.E.) and the 20 model CMIP5 RCP8.5 end-of-the-century discharge projections split between 2050–2074 C.E. and 2075–2099 C.E. We divided the end-of-the-century CMIP5 RCP8.5 discharge simulations into two halves (2050–2074 C.E. and 2075–2099 C.E.) to estimate the sensitivity of our results to different levels of global mean warming relative to the pre-industrial era with continued anthropogenic carbon-dioxide emissions under the RCP8.5 scenario (+3.05 °C by 2050–2074 C.E., and +4.30 °C by 2075–2099 C.E.; Supplementary Fig. 10). The 3.05 °C warming of global mean annual temperatures that we estimate here by 2050–2074 C.E. under RCP8.5 is roughly equivalent to the projected warming that will be achieved by 2099 C.E. under the lower emission RCP4.5 scenario (see Fig. 1 in ref. [61]). Using these datasets, we then calculated the recurrence interval of JAS discharge greater than 48,800 m$^3$/s (Fig. 4b). This discharge

exceedance threshold was equal to JAS discharge in 2007 C.E., the lowest discharge of all instrumental period flood years (Fig. 2b). The recurrence intervals were calculated by sampling 1,000 draws of 30 years with replacement from all datasets (see 'Methods' section).

The median recurrence intervals (in years) were: (i) 4.35 for the instrumental observations between 1956 and 2011 C.E., (ii) 3.57 for the reconstruction between 1956–2004 C.E., (iii) 2.7 for the full reconstruction (1309–2004 C.E.), (iv) 2.5 across all RCP8.5 discharge projections between 2050–2074 C.E., and (v) 2.17 for RCP8.5 discharge projections between 2075 and 2099 C.E. The median recurrence interval for instrumental observations between 1956–2004 C.E. and 1956–1998 C.E. were also 4.35 (Supplementary Fig. 11).

The estimated difference in the recurrence of high discharge greater than 48,800 m$^3$/s between the full reconstruction relative to recent decades, therefore, lies between 24.37% and 37.93%, calculated as (3.57−2.7)/3.57*100 and (4.35−2.7)/4.35*100. As our reconstruction shows a slight wet bias with more recurrent exceedances of the flooding threshold relative to observations over the instrumental period (3.57 years cf. 4.35 years), the 24.37% lower bound of our estimate accounts for this bias explicitly by comparing the full reconstruction only against tree-ring reconstructed instrumental period flows. However, there is substantial overlap in the full distributions of flood hazard recurrence intervals calculated during the instrumental observations (1956–2011 C.E.) and the reconstruction's instrumental period (1956–2004 C.E.) (first two columns in Fig. 4b). This result remains consistent even if we use instrumental observations between 1956–2004 C.E. or 1956–1998 C.E. (Supplementary Fig. 11).

The difference in the recurrence of high discharge greater than 48,800 m$^3$/s between the instrumental data and CMIP5 RCP8.5 in the intervals spanning 2050–2074 C.E. and 2075–2099 C.E. are 42.53% and 50.11%, respectively[62,63]. Therefore, using the reconstruction as a baseline for long-term discharge variability and the CMIP5-simulated discharge as an estimate of climate change impacts on discharge in the basin, we find that recent decades underestimate the frequency of high discharge and in turn flood hazard from natural variability by 24.37–37.93% and climate change impacts by 42.53–50.11%.

In the instrumental observations, mean JAS discharge exceeded 48,800 m$^3$/s in 13 years (Fig. 2b). Despite high discharge during these 13 years, more than half of these years ($n = 7$) experienced no flood. While our recurrence interval analysis focusses on the frequency of high discharge that is associated with the likelihood of flood hazard, many other factors play a role in determining whether high discharge translates to a flood event. These may include rainfall intensity and pattern, landscape heterogeneities, antecedent soil moisture conditions, and land use and forest cover change[64–66]. Our return interval analyses also rely on the assumption that these high discharges will continue to be associated with an increased likelihood of flood hazard in the future, disregarding (for example) potential changes in policy, land use, or infrastructure that may ameliorate 'flood risk'. The occurrence of a flood event that impacts society is however closely intertwined with highly localised human exposure and vulnerability[67,68]. Therefore, our calculations of underestimated high discharge and associated likelihood of flood hazard in the return interval analyses in Fig. 3b only contributes one component of the multiple dimensions of flood risk.

**Climate teleconnections**. We did not find any meaningful or statistically consistent relationship between monsoon season flow in the Brahmaputra River and variance in ocean sea surface

temperatures (SSTs) or indices such as the El Niño-Southern Oscillation (ENSO) or the Indian Ocean Dipole (IOD) (Supplementary Fig. 12). This is consistent with prior studies[15,28,69], even though we used a more up-to-date discharge dataset that extends up through 2011.

## Discussion

We found the magnitude of peak 10-day Brahmaputra River discharge during the JAS monsoon season is tightly coupled to mean discharge for the entire JAS monsoon season. We also show that flood events have almost always occurred during years of high seasonal discharge. Our tree-ring reconstruction of mean JAS Brahmaputra River discharge between 1309 and 2004 C.E. helps inform us about past and long-term hydroclimate variability in this river system. Additionally, the frequency of recurrence of high discharge in the reconstruction relative to the instrumental observations provides us valuable information regarding the likelihood of flood hazard in the region. While the Brahmaputra River has experienced large floods in the past few decades, most notably in 1998 C.E., our reconstruction suggests that the instrumental period that informs our current baseline assessments of flood hazard in the region is actually one of the driest periods over the past seven centuries. This finding and the wet and dry periods we described in our reconstruction are also consistent with other hydroclimate reconstructions in the Southeast Tibetan Plateau covering the upper Brahmaputra watershed[54,55,70,71], and a southward (northward) shifted central Indo-Pacific Intertropical Convergence Zone (ITCZ) during the twentieth century (Little Ice Age, ~1400–1850 C.E.)[72,73]. Climate model simulations under the RCP8.5 scenario suggest wetting over the Brahmaputra River basin leading to increased discharge towards the end of the twenty-first century. While this projected wetting falls within paleo-discharge natural variability estimates, taken in conjunction, the wetter reconstruction and projections relative to the instrumental period suggest that we may be currently underestimating the reconstructed and future frequency of high discharge in the Brahmaputra River watershed.

A limitation of our analyses regarding flood hazard is that we reconstruct Brahmaputra mean JAS monsoon season discharge and not flood years per se. Paleohydrology cross-proxy synthesis between tree-rings and other archives such as geomorphic field stratigraphy[74–77] and speleothems[78–80], the documentation of tree-ring flood-scars that can precisely date past flood events[81–83], and additional tree-ring sampling in the region of traditional[57,84] and non-traditional species[85–87] can help establish more skillful reconstructions of Brahmaputra discharge, its flooding history, and its flooding frequency in future work[88]. Additionally, we focus on the likelihood of high discharge as a proxy for flood hazard, and not on flood exposure and vulnerability[89,90]. In recent years, large advances have been made in the region with accurate flood warnings being made available with lead times of ~8–10 days. Villages taking specific actions have been able to minimise economic and social loss. Therefore, developing such adaptive capabilities to extreme events lends well towards better preparedness in times of increased flood hazard to reduce overall risk[91]. Finally, as lower basin Brahmaputra discharge in Bangladesh is closely tied to upper basin discharge and precipitation, greater availability of real-time river discharge data across all basin states (China, India, Bhutan, and Bangladesh) will help advance these efforts.

## Methods

**Tree ring network**. As an initial selection criterion, we first downloaded tree-ring data located between 20°N–35°N and 86°E–101°E available in the International Tree Ring Databank (ITRDB) that are approximately located within ~670 km or less from the basin boundaries, consistent with spatial autocorrelation in regional

hydroclimate[32]. We then 'standardised'[92,93] each annual raw ring-width series using the signal free (SF) method[94] to reduce the influence of non-climatic growth factors on tree growth and maximize the preservation of common median frequency at decadal to centennial timescales and truncated each chronology to the section with an Expressed Population Signal > 0.85[95,96]. Finally, we retained a tree-ring series as a potential predictor if it correlated significantly with $p < 0.1$ using a 2-tailed t-test with mean JAS flow at Bahadurabad, Bangladesh in its 'raw' and 'pre-whitened' (i.e. serial-autocorrelation removed) forms. We allowed for the inclusion of lag t + 1 predictors in our model, where tree-growth lagged climate and consequently streamflow by 1 year. This is because tree-growth in the current year is often influenced by previous year climate[97]. The details of the series (e.g. species, location, chronology length, lag t + 0 or lag t + 1) retained as predictors in our reconstruction model are described in Supplementary Table 1. The two tree ring series located in Myanmar are developed by us.

**Reconstruction**. We used the Bayesian Linear Regression model prescribed below[37,98] to reconstruct streamflow ($y_t$) in year t, as function of an intercept, slope, and predictor vector X.

$$y_t|\alpha, \beta = \alpha + \beta*X_t + \varepsilon_t$$

with non-informative priors modelled as

$$\alpha \sim N(0, 10^4) \text{ and } \beta \sim N(0, 10^4)$$

The matrix **X** in the equation above contained the principal component scores (PCs—ref.[99]) for all tree ring predictors with eigenvalues greater than 1. We used this Kaiser–Guttman cutoff criteria[100,101] as an estimate of common shared signal (versus noise) between the tree ring predictors. In the reconstruction procedure, we explicitly incorporated the covariance between the streamflow series and the tree-ring series by weighting each tree-ring predictor series by a power of its correlation with the streamflow data during the 1956–1998 C.E. calibration period[102]. This weighting can be expressed by the following equation

$$wTR = uTR*r^p$$

where wTR represents the final correlation-weighted matrix of tree ring chronologies, uTR is the matrix of unweighted tree-ring series normalised to N(0,1) over the calibration period, r is the absolute value of the calibration period Pearson's correlation, and p is a range of exponent powers (0, 0.1, 0.25, 0.5, 0.67, 1.0, 1.5, 2.0)[32,95]. We use this range of powers, as there is no a priori reason for any single correlation weight to be more suitable than any other weight[32,95].

To develop the reconstruction, we chose the 43-year period between 1956 and 1998 C.E. as calibration-validation period to maximize the number of tree-ring predictors available as most tree-ring series ended in 1998 C.E. We used a 'nested' reconstruction approach where we sequentially dropped shorter tree-ring series until the predictor suite was exhausted and developed a new reconstruction model each time a shorter tree-ring series was dropped. We then appended each reconstruction 'nest' together by scaling its variance to the calibration period to develop the longest possible reconstruction of flow possible. Due to the relatively short instrumental period of 43 years (1971 missing) available for calibration-validation, we used a leave-10-out at random calibration-validation approach. In each iteration we calibrated a model on 32 years of streamflow data and validated it on the remaining 10 years. This choice of a 32-year calibration and 10-year validation period provided a trade-off between retaining sufficient years for calibration while developing a conservative estimate of model skill. This is because leave-one-out-cross-validation skill thresholds are easier to pass than leave-k-out validation. For each nest and for each PCA matrix correlation weight, we developed 50 such reconstructions. These 50 leave-10-out-cross-validations when taken together with the 8 different correlation weights used in the PCA analysis gave us 400 reconstructions. The final reconstruction shown in Fig. 3 was calculated as the median of this 400-reconstruction ensemble.

We evaluated the fidelity of our reconstruction using the following metrics: (i) CRSQ (calibration period coefficient of multiple determination), (ii) VRSQ (validation period square of the Pearson correlation), (iii) VRE (validation period reduction of error), and (iv) VCE (validation period coefficient of efficiency) that is equivalent to the Nash-Sutcliffe efficiency test[103]. The full description of these metrics is provided in the supplemental material in Cook et al. ref.[32]. The CRSQ and VRSQ were calculated on the 32-year calibration and 10-year validation periods, respectively.

**Flood hazard and recurrence interval**. The list of flood years prior to the start of the instrumental gauge records in the 1950s at Bahadurabad were primarily collated from Coleman et al. and Chowdhury et al. refs.[10,12]. Flood years from these two studies include 1842, 1858, 1871, 1885, 1892, 1900, 1902, 1906, 1918, and 1922 C.E. Additional records of Brahmaputra floods in 1910 C.E. and in 1787 C.E., when severe heavy rains in the upper basin caused a major flood and the Teesta River changed its course to flow into the Brahmaputra River north of Bahadurabad are from ref.[11].

To calculate the recurrence interval of flood events we computed 1000 bootstrapped draws with replacement of 30 years each from (i) the instrumental observations (1956–2011 C.E.), (ii) the reconstruction over the instrumental period (1956–2004 C.E.), (iii) the full reconstruction period (1309–2004 C.E.), (iv) CMIP5

RCP8.5 2050–2074 C.E. period runoff simulations, and (v) CMIP5 RCP8.5 2075–2099 C.E. period runoff simulations. For the draws from the reconstruction, we used the median reconstruction, while for the CMIP5 data we included all 42 ensemble members of the 20-model suite to represent the full range of variability in the model simulations of discharge. In each draw, we calculated the percentile (P) of the 2007 C.E. JAS discharge of 48,800 $m^3/s$. We then calculated the return interval as 100/(100-P), to give an estimate of the likelihood of the occurrence of high discharge related flood hazard in any given year. For example, if in a random draw of 30 years, the 2007 C.E. flood year discharge placed in the 90th percentile, its return interval probability would be once every 100/(100-90)=10 years. However, if it placed in 80th percentile its return interval would be more frequent at once every 100/(100−80) = 5 years. An alternate approach to the bootstrap sampling we apply here could be to explicitly fit an extreme value distribution to the JAS discharge data.

**Climate models**. We obtained the surface runoff parameter output in the spatial domain upstream of the Bahadurabad gauging station within the Brahmaputra watershed from a 20-model ensemble suite of phase five of the Coupled Model Intercomparison Project (CMIP5, ref. [41]). This was done both for the 1850–2005 C. E. 'historical' simulation period, and 2006–2099 C.E. 'future' simulation period. For the future simulation period, we used the representative concentration pathway 8.5 (RCP8.5) scenario that represents a net radiative imbalance of 8.5 W/m² in earth's radiative budget by the end of the twenty-first century[104]. Supplementary Table 2 lists all CMIP5 models and their respective ensemble members used. We only chose a model run where the same simulation scenario was available both for the historical and RCP8.5 simulation period, to be able to compare 'future' and 'historical' discharge against each other. As a preliminary check, we first tested that the standardised annual hydrographs of the instrumental discharge data and the CMIP5 runoff data matched reasonably well both over the 1956–1998 C.E. calibration period from the CMIP5 historical runs and the RCP8.5 2050–2099 C.E. future simulation period (Supplementary Fig. 13). We did this to ensure that the models accurately capture seasonal runoff dynamics within the watershed, such as the wet JAS monsoon season period and the dry January through April winter period (Fig. 2a). When comparing the variability across model members for the RCP8.5 2050–2099 C.E. simulation period to the instrumental data and the CMIP5 historical data, we found that it showed a large increase in runoff in the months of June through September. Before computing the annual runoff hydrograph using the CMIP5 models and all runoff projections, we bi-linearly interpolated all the CMIP5 runoff data into a common 0.25*0.25 grid, and lagged regions in the upper basin lying in an isochrone greater than 16 days refs. [13,15] by one month to account for the temporal delay between surface runoff in the CMIP5 simulations in the upper part of the basin and the time it would take to reach the Bahadurabad gauging station located in the lower part of the basin (Supplementary Fig. 14). Finally, to develop the CMIP5 estimates of historical and future discharge, we computed z-scores of each runoff simulation using its 1956–1998 C.E. mean and standard deviation, and then scaled these z-scores to discharge (in $m^3/s$) by using the 1956–1998 C.E. instrumental mean and standard deviation. This scaling to the 1956–1998 C.E. instrumental period allowed us to directly compare the reconstructions with the CMIP5 projections, and the magnitude of projected change into the 'future' compared to the 'historical' period and instrumental discharge.

**Superposed epoch analysis (SEA)**. We tested the probability of the association between higher than normal discharge during flood years occurring by chance using a modified double bootstrap SEA[60,105]. To do this, we first calculated the variability in discharge based on 5th, 50th, and 95th percentile of mean discharge across 1000 unique draws of 10 flood years out of the total 16 between 1780 and 2004 C.E. We then determined the statistical significance of this response by comparing its probability distribution to that generated from 10,000 unique draws of 10 years at random ('pseudo-flood years') from the reconstruction between 1780 C.E. and 2004 C.E. We subtracted the five-year pre-event mean across both draws of the 'flood years' and 'pseudo-flood year' to reduce the impact of low-frequency variability on the overall composite mean response. We also conducted SEA using only the 12 pre-1956 C.E. instrumental period in a similar fashion with reconstructed discharge. In this SEA, we used 495 unique event year draws 8 of the 12 flood years due to the lower available sample size.

## Data availability
Tree Ring Data from International Tree Ring Data Bank (ITRDB) [https://www.ncdc. noaa.gov/data-access/paleoclimatology-data/datasets/tree-ring]. Tree ring data are from refs. [32,34,46,106–108]. CMIP5 data [https://esgfnode.llnl.gov/search/cmip]. Huffman, et al. ref.[109] TRMM precipitation data from the IRI Data Library [http://iridl.ldeo.columbia. edu/SOURCES/.NASA/.GES-DAAC/.TRMM_L3/.TRMM_3B42RT/.v7/.daily/. precipitation/]; Harris et al. ref.[51] https://www.CRU data [http://www.cru.uea.ac.uk/ data/]; Schneider et al. ref.[110] GPCC v7 precipitation data [https://www.esrl.noaa.gov/ psd/data/gridded/data.gpcc.html#detail]; https://www.ERA-5[111] runoff—[https://cds. climate.copernicus.eu/cdsapp#!/dataset/reanalysis-era5-land-monthly-means? tab=form]. We downloaded shapefiles for country boundaries from [https://github.com/ nvkelso/natural-earth-vector/] and [https://www.igismap.com/]. Reconstructions and Bangladesh Meteorological Department (BMD) Bahadurabad instrumental discharge

data is available through to the NOAA-NCEI Paleoclimatology Data repository [https:// www.ncdc.noaa.gov/paleo/study/31172]. Any other associated data may also be made available by request from the authors. Source data are provided with this paper.

## Code availability
Our R and JAGS code are available in Supplementary Data 1 along with all 28 standardised tree-ring predictor series, the final reconstruction, and the instrumental discharge data at Bahadurabad.

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

## Acknowledgements
M.P.R. supported by Lamont Climate Center, Lamont Chevron Student Initiative Fund, and National Science Foundation (NSF) Office of Polar Programs (OPP) Arctic Social Sciences award #1737788. E.R.C. by the Lamont Climate & Life fellowship. B.I.C. supported by NASA Modeling, Analysis, and Prediction program, M.P.R., R.D.D., and B.M.B. acknowledge NSF Atmospheric & Geospace Sciences (AGS) award #1303976. D.A.B. supported by the NASA Earth and Space Science Graduate Student Fellowship #80NSSC17K0402. P.J.W. supported by NSF-AGS grant #1638256. M.P.R. thanks Naresh Devineni (CUNY-NY), Dorothy Peteet (Columbia Univ.) Nguyen Tan Thai Hung (SUTD) for helpful discussions. Authors thank Haibo Liu (Columbia Univ.) for assistance with CMIP5 data, Daniel Stahle and Nguyen Trung for helping develop the 2 Myanmar tree ring series. Lamont contribution #8443.

## Author contributions
M.P.R., E.R.C., and B.I.C. designed research. M.P.R. preformed research and wrote the paper with substantial input from E.R.C., B.I.C. and R.D.D. Research ideas were contributed by J.G.P., U.L., C.W., B.M.B, M.U., and D.A.B. J.J. and P.J.W. provided the streamflow data and assisted in its interpretation. All authors edited the manuscript.

## Competing interests
The authors declare no competing interests.
