## [Peer Review File · Nature Communications]

Reviewer #1 (Remarks to the Author):

NCOMMS-20-04754-T

"Six centuries of reconstructed Brahmaputra River flow demonstrate underestimated flood risk?"

Submitted to Nature Communications by Rao et al.

The manuscript presents a tree-ring reconstruction of monsoon season discharge to investigate the flood dynamics in the Brahmaputra River basin over 1417-2004. This is a significant piece of work. However, improvements are needed before the paper can be considered acceptable in Nature Communications.

Although this work represents an incremental contribution to this important topic, it seems there isn't novelty in term of methodology. The authors, by the way, have published a similar work (see Rao et al. 2018: "Six Centuries of Upper Indus Basin Streamflow Variability and Its Climatic Drivers") with the same regression model and the same variable (discharge). Hence, its novelty is in the insights it provides. It has essentially two messages: 1) the 1956-1989 period (instrumental) baseline of flood risk is the second driest period over the last six centuries; and 2) the tree-ring reconstruction suggests that the flood risk estimated from the instrumental record is severely underestimated, independent of the future changes (RCP8.5).

However, the R^2 between the observations and the reconstruction is about only 50% (fig 3a). This implies some missing significant contributions of other drivers controlling the tree-ring growth (e.g. the sun radiation?). Hence, the conclusion that the 1956-1989 period is the second driest can be biased because the previous minima/maxima in the tree-ring records could be created by other processes than just the monsoon rainfall. The impact of these missing factors should be evaluated in the paper before concluding about the 1956-1989 period.

Specific comments:

The validity of t-student statistics applied (line 151-153) should be justified explicitly: e.g. are the variance of tree-ring reconstruction and that of instrumental discharge equal?

Standardization (signal free method) and filtering (pre-whitening) procedures are applied to the tree ring dataset (see lines 300-305), but nothing is said, or not clearly expressed, regarding the homogenization of the Bahadurabad discharge record. It is surprising that the Pearson correlations between each tree-ring site and JAS flow are, though very low (see Table S1 0.26-0.53, and 2 negatives). Is the use of linear relationship between JAS discharge and tree-ring records justified? It would be helpful to display scatterplots of the JAS discharge versus tree-ring records in the supplementary.

The authors reconstruct the Brahmaputra discharge, that is a response to a large-scale common hydroclimate mode representing the Brahmaputra watershed (and even more from China to Myanmar). At this scale it is most likely that their tree-ring reconstruction contains a signal from the entire Ganges-Brahmaputra-Meghna hydrological system and not only from Brahmaputra. Perhaps, this could explain also the low correlation and small R^2 between the observed discharge record and the tree-ring reconstruction. It would be interesting to compare these results, as well, with the long-

term daily discharge data available for the Ganges at Hardinge Bridge in Bangladesh (more than 60 years, available from the Bangladesh Water Development Board). In the same way, the authors list the Brahmaputra historical flood years from ref 10 and 12; but these events correspond to Ganges-Brahmaputra-Meghna flood or Bangladesh flood years.

The authors have tested the sensitivity of their reconstruction by excluding 5, the most distant, of the 16 predictors (tree-ring) and they concluded that the reconstruction is not overly sensitive to the inclusion of the more distant predictors. So, why did the authors consider in their model 16 predictors if 11 is enough? Can the authors explain their choice? Did the authors test other predictors combinations? How are the inter-tree-ring record correlations? What is the effect of redundant predictors (if available) on the discharge reconstruction and their uncertainty?

Fig S7. The authors said " The comparison illustrates that the multi-decadal dry and wet periods over the basin suggested by our JAS reconstruction are also suggested by larger scale reconstructions of spatial drought variability." It's not clear, the corr coeff. is very low $r=0.27$, what can be really learned from this comparison? A low-pass filtering form may bring more information.

On which basis has the period 1417-2004 been divided into sub-periods of 10 years? Why 10 years? It would be helpful to know how these findings will be change/robust into others time size. Did the authors check the stationarity of the discharge reconstruction? Do the last 50 years of the discharge reconstruction follow the same probability distribution of the 600 years?

Line 354: The authors estimate the "recurrence interval of flood events" using 1000 bootstrap values, i.e random sampling with replacement from 30 years of each dataset (instrumental observations, instrumental reconstructions, full reconstruction and CMIP5). Please detail more clearly the procedure used.

The absence of relationship between the monsoon seasonal flow and the modes of climate variability as ENSO and IOD looks surprising and should be detailed and discussed more: what analysis was performed to claim this finding?

line 72-73 : There is no information about the Bahadurabad discharge record (provider, availability, precision, quality, sample,...)

line 131-132 : There is no information about the regional precipitation dataset used. How and why were TRMM, GPCC and CRU selected and used?

Fig1: please give the signification of correlation coefficients

Line 143 : correct "that that"

Reviewer #2 (Remarks to the Author):

I have completed my review of the manuscript (NCOMMS-20-04754-T) "Six centuries of reconstructed Brahmaputra River flow demonstrate underestimated flood risk" by Rao et al. I found

the manuscript to be fairly straightforward and well written. This is interesting and important research for our discipline, and I have no doubt that this manuscript will be a useful addition to the literature. I offer discussion of a couple of issues for consideration.

There are essentially two main arguments in this manuscript; 1) that the tree-ring records are faithful proxies of monsoon season river discharge and 2) that the estimates of seasonal discharge derived from the tree-ring records provide reliable estimates of “flood risk”.

I believe that the analyses described in the manuscript make it clear that the discharge reconstruction is statistically sound overall and does a good job of estimating the mean discharge. And certainly seasonal discharge is related to flood risk. However, in my opinion the manuscript somewhat overstates the usefulness of the reconstruction to provide reliable estimation of changes in “flood risk”. From my perspective this is really the only serious potential issue with this manuscript so I will discuss my overall thoughts on this matter first followed by some specific questions and suggestions and then a few relatively minor items (many of these can also be found on an attached “track changes” version of the submitted manuscript).

On line 103 the authors describe how seasonal discharge and maximum 10-day mean discharge (essentially a proxy of major flooding) are coupled, which is not surprising, but (here I disagree with the authors’ perspective) the relationship is not very strong (less than half the variance is explained). The essential argument of the manuscript is then given in the Conclusions (line 254)

“We found the magnitude of peak discharge during floods in the JAS monsoon season to be tightly coupled to mean discharge for the entire JAS monsoon season. This allows us to use our tree ring based reconstruction to inform our understanding about both past and long-term variability in flood risk in the watershed.”

To rephrase it less elegantly- because mean seasonal discharge is “coupled” with maximum 10-day mean discharge, and the tree-ring proxies can explain 53% of the variance in seasonal discharge, the reconstruction is a good proxy of “flood risk”.

So my questions about this argument are

- 1) If estimating “flood risk” is the goal, why not reconstruct maximum 10-day discharge? It seems like it would be a lot less convoluted.
- 2) Figure 3a, and especially 3b indicates that there are many years when the reconstructed discharge exceeds the value (48.8k m³) indicating a “flood”, but no flood is recorded and there are years when there are documented floods but reconstructed discharge values are relatively low. I am curious to know what the actual number of “hits/misses” are. The authors deal with this to an extent in their SEA discussion, which I think was well done. However the SEA approaches the issue from a mean value perspective, which I think tends to blur the actual efficacy of the model. In any case I think it is important that the authors at least touch on the fact that there are many instances where the model falsely indicates what the authors have defined as “floods”, or fails to indicate a recorded flood.
- 3) Another argument made by the authors is that the instrumental period (1956-2016) (and especially the early portion 1956-1989) has been unusually dry compared to the full reconstruction (e.g. see line 150). This argument is primarily supported by a return interval analyses (RIA) that shows that the RI of “floods” is nearly twice as long in the instrumental period as compared to the overall reconstruction. Looking at Fig. 3a one can see that there are two pronounced low-flow

periods between 2004-2016 so I think this argument would be strengthened if readers are provided the value for the 1956-2004 instrumental mean. Since this is the period in common with the reconstruction, I believe readers would like to know how different it might be from the 1956-2016 mean. Also we are told in the Methods that the RI was performed on the 1956-2004 period but in the Results (line 211) the 1956-2016 period is indicated. So it is not clear whether the period used for the RI is problematic.

I think this is really the only overall issue with the paper. I think as a streamflow reconstruction, there are really no problems, but as a proxy for flood risk I don't think it is as persuasive as it could be.

Following are some specific notes. Other more minor items can be found on the track changes doc.

Lines 103-107 "closely coupled" and "strongly associated" might be overstating the relationship. We're talking about less than 50% of the variance. Also I think it would be helpful to state whether the discharge data described here and in Fig 2b. are the instrumental or reconstructed. I assume instrumental since values after 2004 are given but it would be help to be specific. It could be useful to also show the relationship with reconstructed discharge.

Lines 108-117 I'm not sure I understand how this discussion advances the argument that the recon is a good proxy for flooding. I assume the point is that flood years usually co-occur with lengthy periods of above average flow. However examination of Fig.S3 appears to show that it is very common for above average flows of long duration to occur in years when no flood was recorded.

Lines 135-147 I don't find this necessary, but I suppose someone might question the more distant chrons being included. I think this could easily be moved to the Suppl Mat.

Lines 150-152 I am curious why the full instrumental period mean 1956-2016 is used when the recon only extends to 2004? Maybe it doesn't make any difference, but since the case is being made that the instrumental period is lower than the recon I think readers would want to know how the means compare over the common period. I think this is especially relevant since the 1956-2004 period is used in the other analyses.

Line 211 The authors state "we used the instrumental data (1956-2016),..." Is this correct? It says 1956-2004 in Methods

Line 401 Using flood data as event years in an SEA of discharge seems problematic given that they are not entirely independent, however, I think the authors have done a really good job of estimating the true significance of the difference in means of the flood years. I am curious however whether using the values (upper) for the confidence intervals for the draws composing the 'pseudo-flood years', would still result in a significant difference.

Line 289 Data Availability- I think that each of the 16 tree-rings sites used in the analyses needs to be cited in the Refs. or Supp. Mat refs. Also I think that if reproducibility is important the supp mat or the ITRDB archived materials need to be much more specific. For example, not only the full reconstruction but each model run as well as the calibration and verification stats for each model, the sample size (by year) for each model etc. should be in the SM or archived at the ITRDB.

Figure 3a The recon seems to do a poor job on the extremes and especially the dry extremes (which seems unusual). It might be useful to apply the bias correction technique discussed by Robeson et al. (2019).

Figure S8. Table S3. There are instances where the term “cyclone” or “cyclonic” are used but it is not specifically noted as a “tropical cyclone”, which I think would be better.

Table S1. Is there any order to these sites? It might be helpful to list them by Lat and Long so that readers can get an idea of where individual sites are on the map.

Please let me know if I may offer any clarification or be of any other assistance.

Sincerely,

Matthew Therrell
Professor

Refs

Robeson, S. M., Maxwell, J. T., & Ficklin, D. L. (2020). Bias correction of paleoclimatic reconstructions: A new look at 1,200+ years of Upper Colorado River flow. *Geophysical Research Letters*, 47, e2019GL086689. <https://doi.org/10.1029/2019GL086689>

Reviewer #3 (Remarks to the Author):

The authors presented a study reconstructing the flow of the Brahmaputra river based on tree-rings over the last six centuries. The study conclude that “flood risk is underestimated by the instrumental record”, and that “recent observations to will be insufficient to determine the degree of flood risk for the coming decades”. The importance of such studies supporting this is high, mostly due to the large impact that floods have on millions of people living in the floodplains of this river, but also to support one more time, that systematic records are not enough to determine properly flood hazard. Yet the manuscript provided by Rao et al., fails under my opinion is the following items, which I consider major problems, specially point 1 and 2, preventing me, unfortunately, to accept this ms:

- 1) Authors are mixing concepts between flood hazard and flood risk along the entire ms. This imply that the ms is not technically sound, since not vulnerability, neighed exposure assessment is provided. Similarly, there are some other concept in the ms used that are somehow not commonly used in hydrology as, climatological flow (although this is less importance)
- 2) Authors present “flood risk” from now I consider it as “flood hazard” without analyzing stationarity and providing a way to compare the results with classical flood hazard assessment, which normally use the concept of return period. The stationarity, highly discussed in hydrology and paleohydrology, is not mentioned neither.
- 3) The outcomes are based on a relationship described by the 53% explained variance. I am not sure if we can call this as strong evidence.

- 4) Results could be novel, but the general conclusion about the drawbacks of using short records is not.
- 5) I am missing key references to palhydrology in the region, as well as their inclusion in the discussion related to wet/dry periods.
- 6) The methods here used are broadly accepted by the tree-ring community, but an identification of the assumption, and including a discussion of the limitations of the results is missing in the ms. Thus, I should accept that reconstructing with the 53% of explained variance is considered ok, as I did not see any discussion related to potential drawbacks or limitations. A interval confidence level should be also provided in the text, and not only graphically masked in the Figure 3 (gray bars).
- 7) Only a RCP is used, without explain why this has been chosen. I imagine they are interested in only one extreme of expected impacts...

Other comments are below:

The title is not technically sound. Authors talk about flood risk, but not risk assessment is provide. It seems they are unaware about the risk concept (as broadly defined, last GAR 2019), which must include an assessment of the vulnerability, exposure and hazard. Here authors don't show any assessment on exposure and vulnerability, and partially on hazard.

Line 18: vulnerable? You mean exposed? Vulnerable includes how people is affected by any impact, in relation to its cultural or socio-economic characteristics (among others). And I guess is defined to people (or assets) living (being) in floodplains rather than floodplains itself, to which I guess it is better to use "recurrently affected by"

Line 20-23: This sentence should be split in two, one talking on intensified summer monsoon and other in flood risk. In fact, the intensified monsoon is not the only component leading an increase on flood risk.

Line 24: Brahmaputra river

Line 24: mean JAS flow discharge.

Lines 26-27: these findings are not novel, but expected, true?

Line 27-29: how these findings related with the existing paleofloods records?

Line 30: you are not providing flood risk. This is not technically sounds. The term "flood risk" is not well used in the entire manuscript.

Line 33-35: the philosophy here expressed "a focus on recent observations to evaluate flooding in the basin will be insufficient to determine the degree of flood risk" is not novel. In fact, statisticians and paleohydrologist have broadly stated the same idea since 1960's. From the sentence, it seems that you ending with a general conclusion. I'd suggest here to highlight your own conclusions otherwise, it looks not novelty for me.

Line 45-50: May be it is a good idea to short this sentence or to list the items for a better read.

Line 54-56: it sounds repetitive. The idea is already introduced in the lines 41-42.

Line 69: I have read this paper and I did not found where is indicated that "flood risk" will increase in Brahmaputra. This is for me a major problem of the ms so far, I think you are missing concepts between flood hazard and flood risk. May be you assume that an increase in flood hazard results in an increase in flood risk, but technically is not sound.

Line 79-76: I highly miss the existing bibliography on paleohydrology and historical hydrology here. For instance, Puni and Ravishanker (1983), contextualizing extreme flood event into long-term historical records. Kale et al., (1987), who suggested an increase on floods during last decades...or

Kale et al., (1997, 2000) suggesting clustering of large flood events in the recent decades. In other part of India there are other works, like Ely et al., (1996) or Thomas et al., (2007). I guess some of these works, especially in the Eastern part of India should be mentioned? How these findings relates to yours?

Line 77: using tree-rings as a surrogate of hydroclimate is a way to indirectly reconstruct "streamflow". I miss here references that use tree-rings records to date and reconstruct individual past flood events. Some of them are: Balesteros-Canovas et al., 2015 or Wilhelm et al., 2019.

Line 88: you mean between monsoon and seasonal flow discharge.

Line 85: why you use only RCP 8.5?

Line 86: I strongly disagree on the point: how flood risk in recent decades compares to longer-term estimates over the last several centuries. For the points indicated about the concept "risk".

Line 94: this information is mentioned already. Please don't repeat information.

Line 98-100: This sounds as a strong assumption. Do you have information supporting this? I don't have doubts about the relationship between tree-ring widths and temperature or even precipitation, although this provides a more complex figure. Translate this relation into streamflow would be ok, but I consider as an indirect relationship, the problem I see here is how to link exactly tree-ring width with maximum flood during 10-days. Do you have like a kind of model-based results to support how seasonal precipitation is transformed into a hydrography signal at the studied reach river? Or other kind of information. Otherwise, for me it is an assumption and you should treat it (and presented) properly in the ms.

Line 102: did you probe other combination of days? I imagine your above-mentioned assumption is based on this relationship but, did you test others combination of mean- hydrographs values?

Line 103-104: this is obvious.

Line 113: what it is climatological flow? You mean ordinary flow? Bankfull discharge? I not sure if climatological flow is a broadly accepted term in hydrology.

Line 122: do you think that 42 years is enough to calibrate and validate the relationship that then is used to reconstruct six centuries?

Line 128 -130: Actually, this has more sense, as tree growths are physically connected to the moisture in soil provided, mostly, by precipitation. I think this should be appear first and then admit that, as consequence, an assumption about tree-ring widths and seasonal streamflow can be done. Providing here the correlations could be interesting.

Line 131: I am aware about tree-ring climate relationship, and see that 53% is ok, but at the end, it means that the reconstructed flow discharge could be or could not be...This should be discussed.

Line 147: provide interval confidence data!

Line 171: how reliable is your reconst. in this time i.e. 1435? I miss information about the reliability (EPS) of the used tree-ring chronologies.

Line 201: I can not understand how you estimate flood risk relative to observational period. I assume that you are simply wrong, and instead you should have used the term "hazard". In this case, you should reconsider your text and correct it. But the repetitive use of the term "risk" along the ms, make me think that you are really mixing concepts, and this is more problematic. You cannot talk about risk, at least, you are assessing the vulnerability and exposure of the population living in the floodplains of Brahmaputra.

Moreover, flood hazard is commonly based (in hydrology) in the concept of return periods, to which a large discussion about stationary has been risen last decades. You can have a look in a recent recompilation paper where paleo hydrology is taken into account for flood hazard assessment, and where all these concepts are treated (Wilhelm et al., 2019). How your data should be compared with other results using the concept of return period? Could you provide this information?

Why you used only RCP 8.5? and not the entire set, or at least the two most extreme, scenarios

Line 228-229: it is a bit strange to read here that tropical cyclones are major cause of floods. May be this can be merged in the information provided in line 41-42.

Line 233-235: already mentioned and it is not novelty.

Line 240. Then, you used this dataset for your tree-ring / streamflow relationship?

Line 227: I miss a discussion with the long-term climate-flood linkages provided by paleohydrologist in the region.

Line 279: data availability. You should include the scripts you used, and specify which computational language you used.

Line 343: recurrence interval. This way to treat the flow data is different than the one normally accepted in hydrology. Why did you used this way, which advantages has in relation to the classical one, should be considered as a novel approach?

Reviewer #1 (Remarks to the Author):

NCOMMS-20-04754-T

"Six centuries of reconstructed Brahmaputra River flow demonstrate underestimated flood risk?"

Submitted to Nature Communications by Rao et al.

The manuscript presents a tree-ring reconstruction of monsoon season discharge to investigate the flood dynamics in the Brahmaputra River basin over 1417-2004. This is a significant piece of work. However, improvements are needed before the paper can be considered acceptable in Nature Communications.

Although this work represents an incremental contribution to this important topic, it seems there isn't novelty in term of methodology. The authors, by the way, have published a similar work (see Rao et al. 2018: "Six Centuries of Upper Indus Basin Streamflow Variability and Its Climatic Drivers") with the same regression model and the same variable (discharge). Hence, its novelty is in the insights it provides. It has essentially two messages: 1) the 1956-1989 period (instrumental) baseline of flood risk is the second driest period over the last six centuries; and 2) the tree-ring reconstruction suggests that the flood risk estimated from the instrumental record is severely underestimated, independent of the future changes (RCP8.5).

The Brahmaputra and Indus rivers are controlled by very different climate mechanisms (monsoon precipitation vs winter seasonal snow). The Brahmaputra River is also one of the largest in the world in terms of its mean annual discharge. Currently there are no long-term annually-resolved reconstructions of Brahmaputra River discharge, leaving a large gap in our understanding of the paleohydrology of the region. Additionally, floods in the Brahmaputra basin are more frequent and have a much greater humanitarian impact than in the Indus River where drought is more commonly the predominant concern. Therefore, these two works are completely independent with little relationship to each other in terms of the scientific insights provided by them.

R1.1 However, the R^2 between the observations and the reconstruction is about only 50% (fig 3a). This implies some missing significant contributions of other drivers controlling the tree-ring growth (e.g. the sun radiation?). Hence, the conclusion that the 1956-1989 period is the second driest can be biased because the previous minima/maxima in the tree-ring records could be created by other processes than just the monsoon rainfall. The impact of these missing factors should be evaluated in the paper before concluding about the 1956-1989 period.

We use the trees-ring records as a proxy to estimate discharge. The growth of each individual tree, the site-level tree-ring chronology, and the full pool of 28 candidate tree-ring predictors are certainly influenced by factors other than just those that also influence discharge. In the reply to the comment R1.4 we document how we use Principal Components Analysis (PCA) to first isolate common ‘shared variance’ between all the predictors. In addition, discharge is also influenced by factors other than climate, and more specifically for our case climate information captured by the ‘shared variance’ between tree-ring predictors. Therefore, uncertainties in any reconstruction are to be expected.

The uncertainty bands in the reconstruction ensemble reflect the range of variations that are due to other factors (i.e. ‘unexplained variance’). If for example, we consider a case in which the tree rings are totally non-informative, and there would be no temporal structure in the error residuals derived from model predictions. In this case, the paleo reconstruction would look exactly as the historical record in terms of its ensemble uncertainty and return period estimates with each historical year being equally likely. As the tree rings become more informative, persistent periods that were anomalous would be identified and the degree of the anomaly would also be reflected by the uncertainty ensemble, which will shrink as the level of proxy information increases. Consequently, discriminating between persistent wet and dry periods becomes feasible, with a corresponding probability. The VRE and VCE statistics of our model indicate that we better estimate ‘observed’ discharge using the tree-ring predictors as opposed to a guess of the instrumental mean as the discharge for each year.

The question regarding whether the first principles assumption that monsoon rainfall is important for both past discharge and past tree growth is harder to evaluate. Though we note that because we use a pool of predictors, we are more concerned as to whether the ‘shared variance’ of predictors is influenced by past hydroclimate. At the latitudes where our predictors are located (20-35°N), light limitation is unlikely at the individual site level, and as a common explanatory variable for all 28 tree-ring predictors. Most of the tree-ring sites used as predictors are not altitudinal tree-line sites either. However, temperature could still indirectly play a role in controlling regional tree-growth as weak monsoon years are also warm years due to lower cloudiness. Warmer temperatures (in conjunction with lower precipitation) could further lower discharge by lowering the ‘runoff efficiency’ due to evaporative losses with a high VPD. Nonetheless, even in this case, where lower precipitation causes warmer temperatures, we expect their influences on trees-growth and discharge to be in the same direction (i.e. negative). Finally, our results in Fig. 7 (main text Fig. 1b) show that the PC1 timeseries of the 28 tree ring predictors that explains around 24% of the shared variance between all the tree-ring predictors correlates with regional JAS precipitation ($r = 0.3$ to 0.65).

Also see our response to Editor Comment E1 regarding improved model R^2 values.

Specific comments:

RI.2 *The validity of t-student statistics applied (line 151-153) should be justified explicitly: e.g. are the variance of tree-ring reconstruction and that of instrumental discharge equal?*

The reconstruction is calibrated to the mean and variance of the instrumental data between 1956-1999. Therefore, the variances over the calibration period are equal by design. We modified the text here to state that more clearly and specify that the comparison is specifically for the full reconstruction (1309-2004) against instrumental discharge (1956-2011).

Modified lines 149-163: *“The reconstructed discharge estimates during the observational period are shown in Fig. 3a and our full reconstruction between 1309-2004 C.E. is presented in Fig. 3b. While the reconstruction is calibrated to the instrumental mean and variance in the reconstruction procedure (Fig. 3 c-d), we found that the mean reconstructed discharge over the full reconstructed period between 1309-2004 C.E. was significantly higher than the instrumental mean between 1956-2011 C.E. ($46,993 \pm 812 \text{ m}^3/\text{s}$ cf. $43,350 \text{ m}^3/\text{s}$, difference of means = $3,644 \text{ m}^3/\text{s}$, t-statistic = 5.11, $p < 0.01$) (Fig. 3a). The uncertainty range around the mean was derived from the 5th and 95th percentiles of the means across all 400 iterations of the median reconstruction. The reconstruction was also significantly ($p < 0.01$) wetter than the instrumental period even if we used the 1956-2004 C.E. instrumental mean of $43,442 \text{ m}^3/\text{s}$ or the 1956-1998 C.E. calibration-validation period instrumental mean of $43,233 \text{ m}^3/\text{s}$. This difference between the instrumental and reconstructed mean is largely driven by the first three decades of the instrumental observations between 1956-1986 C.E. that was unusually dry in the long-term context of the past seven centuries⁹. The instrumental mean between 1956-1986 C.E. was $41,206 \text{ m}^3/\text{s}$.”*

As mentioned in the statement above, we now include a comparison of the kernel density profiles of the instrumental observations and the reconstructions between 1956-1998 in Fig 3c and 3d (in the main text). The updated figure is shown below.

Fig. 3. Updated Fig. 3c and 3d in the main text, describing the kernel density profile of mean JAS instrumental discharge (1956-2011; in red), reconstructed discharge (1309-2004; in green); and CMIP5 end of the century simulated discharge (2050-2099; in green). The means of the 3 datasets are shown as horizontal lines, along with the mean instrumental flood year discharge (in purple). The inset figure (3d) shows the kernel density profiles of mean JAS instrumental discharge (in red) and reconstructed mean JAS discharge (in blue) over the calibration-validation period (1956-1998) along with their means. This illustrates the reconstruction well captures the features of instrumental discharge in this period.

R1.3 Standardization (signal free method) and filtering (pre-whitening) procedures are applied to the tree ring dataset (see lines 300-305), but nothing is said, or not clearly expressed, regarding the homogenization of the Bahadurabad discharge record.

Overall, the Brahmaputra discharge data at the Bahadurabad gauge is of pretty high quality¹⁹, and has been used in its original form in numerous prior studies^{20,21}. Therefore no ‘homogenisation’ is applied to the gauge data. We developed our monthly estimates of discharge using the daily data. In our updated calculation we set the monthly discharge of a month to be missing if it had more than one-third missing days in the month. Due to a large number of missing days after 2011 we truncated the discharge data to extend from 1956-2011. As an independent comparison of whether discharge at Bahadurabad matches runoff simulated by the ERA-5 reanalysis, below we computed the spatial correlation between JAS discharge at Bahadurabad and JAS ERA-5 runoff (Fig. 4, a &c). Below the spatial correlation plots we also present timeseries comparisons between Bahadurabad discharge and ERA-5 simulated discharge averaged between in the grey shaded box in Fig. 4, b & d. The extremely high correspondence ($r=0.80$ for raw data, and $r=0.90$ for first-differenced data, 1981-2011, 31 years) suggests that the gauged discharge record at Bahadurabad is reliable. This figure is now included as Supplementary Fig. S3 in our manuscript.

Fig. 4 Pearson correlation between mean JAS discharge at Bahadurabad, Bangladesh and ERA-5 reanalysis modelled JAS Runoff between 1981-2011 (31 years). Correlations are calculated using discharge and modelled runoff data (**a & b**), and using first-differenced discharge and runoff data (**c & d**). For the timeseries comparisons in **c** and **d** ERA-5 modelled runoff was averaged in the grey shaded box shown in **a** and **b** spanning the upper basin. ERA-5 runoff is modelled as the sum of surface and sub-ground runoff and is driven primarily by precipitation, melting snow, and soil storage in the model formulation. The high correlations between discharge at Bahadurabad and independent estimates of runoff from a hydrologic model driven using climate variables validates the robustness of the discharge data and that lower basin discharge at Bahadurabad is driven by upper basin runoff (that is in turn controlled by upper basin precipitation - see Main Text Fig. 1 and Supplementary Fig. S2). Only correlations significant at $p < 0.05$ using a 2-sided t-test are shown.

As an additional comparison, we compared the Bahadurabad gauge discharge data against the Global Flood Detection System (GFDS) Site 27 data. The GFDS Dartmouth Flood Observatory gauge is located around 100 km upstream near the intersection of the Brahmaputra and Teesta rivers. The GFDS data is available from 1998-present. Over the period of overlap, mean JAS discharge at the two gauges are correlated at $r=0.71$ ($p<0.01$, 1998-2011). The GFDS Site 27 mean JAS discharge is also significantly correlated with ERA-5 simulated mean JAS runoff, $r=0.60$ ($p<0.01$, 1998-2019). This comparison is shown below in Fig. 5, but is not included the manuscript.

Fig. 5 Comparison of mean JAS discharge at the Bahadurabad gauge (1956-2011) and mean JAS at Global Flood Detection System Site 27 located approximately 100km upstream (in red). The timeseries in blue is the ERA-5 upper basin mean runoff. Bahadurabad and Site 27 data have a Pearson correlation of $r=0.71$ ($p<0.01$, 1998-2011). GFDS Data is available at: <http://floodobservatory.colorado.edu/SiteDisplays/20.htm>

***R1.4** It is surprising that the Pearson correlations between each tree-ring site and JAS flow are, though very low (see Table S1 0.26-0.53, and 2 negatives). Is the use of linear relationship between JAS discharge and tree-ring records justified? It would be helpful to display scatterplots of the JAS discharge versus tree-ring records in the supplementary.*

We present the scatterplots below in Fig. 6. In our reconstruction procedure we do not use the tree-ring records directly to reconstruct JAS discharge. Instead we use the leading Principal Component (PC) timeseries of the predictors in each nest to reconstruct discharge. This procedure helps better distill common shared variance between all the predictors that is driven primarily by hydroclimate variability in the region. In our nested reconstruction procedure, we successively drop the youngest tree-ring predictor series to develop the longest reconstruction of discharge possible. While we retain PC series with eigenvalues greater than 1 as potential predictors we found that the PC1 of each nest consistently showed the highest correlations with JAS discharge. PC2 and PC3 also often showed high correlations, however their signs often flipped as commonly occurs in Principal Components Analysis.

Across the 11 reconstruction nests:

- i. Pearson correlation between PC1 and JAS instrumental discharge ranged from 0.51 to 0.73 (median=0.65)
- ii. The median *absolute* correlation between PC2 and JAS instrumental discharge was: 0.35 . The 2 highest correlations between PC3 and JAS instrumental discharge were -0.40 and 0.33. The maximum and minimum number of predictors used in our reconstruction are 28 and 10 respectively. Fig. 6 shows the scatter plot between JAS instrumental discharge and the PC1 and PC2 timeseries for these 2 end member reconstruction nests and for 2 intermediate nests.

Fig. 6 Scatterplot between mean JAS Brahmaputra discharge at Bahadurabad between 1956-1998 and the first two principal component (PC) timeseries obtained from the principal components analysis (PCA) of 28, 23, 17, and 10 tree ring predictors used in the reconstruction (in blue and red respectively). The maximum and minimum number of tree ring predictors used in the reconstruction are 28 and 10. The figure also presents the Pearson correlation between each PC timeseries and instrumental discharge.

In our manuscript in Fig. 1b of the main text we include the spatial correlation between the PC1 timeseries of all the 28 tree ring predictors and JAS precipitation (see Fig. 7 below). The region of positive correlation between the PC1 timeseries and extends across most of the Brahmaputra watershed. This result shows that the predictors chosen reliably capture regional precipitation independent to their relationship with Brahmaputra discharge. We also include our R code for users who may wish to evaluate this relationship between the predictors and instrumental discharge. Therefore, we currently don't include the scatterplots presented above in Fig. 6 in the supplementary text of our manuscript.

Fig. 7 Spatial field correlation between the first principal component (PC1) of the 28 tree ring predictors (variance explained: 24.86%) and mean JAS precipitation (1956-1998). Spatial correlation is against CRU Ts 4.01 precipitation. Only correlations significant at $p < 0.05$ using a 2-tailed t-test are shown. Supplementary Table 1 describes the loading of each tree-ring series on PC1.

RI.5 The authors reconstruct the Brahmaputra discharge, that is a response to a large-scale common hydroclimate mode representing the Brahmaputra watershed (and even more from China to Myanmar). At this scale it is most likely that their tree-ring reconstruction contains a signal from the entire Ganges-Brahmaputra-Meghna hydrological system and not only from Brahmaputra. Perhaps, this could explain also the low correlation and small R^2 between the observed discharge record and the tree-ring reconstruction. It would be interesting to compare these results, as well, with the long-term daily discharge data available for the Ganges at Hardinge Bridge in Bangladesh (more than 60 years, available from the Bangladesh Water Development Board). In the same way, the authors list the Brahmaputra historical flood years from ref 10 and 12; but these events correspond to Ganges-Brahmaputra-Meghna flood or Bangladesh flood years.

The Brahmaputra is largely a free-flowing river²² though there are plans underway to build a dam on the Yarlung Zangbo section of the Brahmaputra in China. Tree-ring reconstructions of discharge are only possible for natural free flowing rivers. This is because dams can partially decouple climate variability and discharge. Though we note that given enough years of ‘natural flow’ data tree-ring reconstructions can be used as a means to estimate flow that would have occurred in the absence of human built reservoirs⁶.

The Ganga has hundreds of dams including one at the Farakka Barrage built by India the 1970s that diverts its water away from Bangladesh. Consequently, there are no estimates of natural flow of the Ganga from gauge records, and the Ganga is unfortunately far from being a free-flowing river²². Co-authors of this study have in previous analyses evaluated short-term flood forecasts of the Brahmaputra and Ganges rivers together based in antecedent precipitation^{20,21}. We are hopeful that the limitation placed by the absence of ‘natural’ discharge data may partially be resolved by the advances in river runoff reanalysis products such as ERA-5 (see Fig. 4), river discharge

reanalysis such as the Global Flood Awareness System (ref. ²³), and satellite altimetry products²⁴ that will likely become available soon. However, this is beyond the scope of analysis presented in our paper currently.

We agree with the reviewer that the tree-ring signal reflects hydroclimate variability in a much broader region. The spatial correlations above in Fig. 7 suggest that the shared variance between the tree-ring predictors is influenced by precipitation in eastern India, Bhutan, and southern Tibet, China. The spatial correlations also hint at a negative correlation between ‘mainland India’ precipitation and precipitation in this region (also see reply to Reviewer 1, comment 1.10). This relationship is similar that between mean JAS Bahadurabad instrumental discharge and JAS precipitation shown in Fig. 1 of the main text.

We tried our best to only compile flood events that specifically focus on or mention that the Brahmaputra River. Though as the reviewer says the two river systems may flood together and the categorisation of some events may be unclear. This might also explain why not all flood years compiled by us are ‘wet’ in our discharge reconstruction presented in the main text Fig. 3. We include this as a limitation of our study in our discussion regarding ‘Historical Flood Events’.

Also see reviewer comment R2.9.

Lines 197-205: *“However, we found this relationship to be much weaker in the reconstructions. Six of the 12 pre-1956 C.E. flood events occurred in relatively dry years. SEA of the 12 flood years prior to 1956 did show a median result of wet conditions in the year in which a flood was documented (Supplementary Fig. S7). However, this result was not statistically significant at $p < 0.05$ despite the 95th percentile of bootstrapped discharge responses being significant at $p < 0.001$. This asymmetric response may be partly due to some of these historical flood years being undivided Bengal (Bangladesh, and West Bengal, India) flood years and not solely Brahmaputra flood years, and lack of information regarding the magnitude of flooding in the historical sources that we used.”*

RI.6 *The authors have tested the sensitivity of their reconstruction by excluding 5, the most distant, of the 16 predictors (tree-ring) and they concluded that the reconstruction is not overly sensitive to the inclusion of the more distant predictors. So, why did the authors consider in their model 16 predictors if 11 is enough? Can the authors explain their choice? Did the authors test other predictors combinations? How are the inter-tree-ring record correlations? What is the effect of redundant predictors (if available) on the discharge reconstruction and their uncertainty?*

We wanted to show that the inclusion of these distant predictors added model skill. We evaluated this using the model validation statistics, where the model performed better against data it did not ‘see’ during calibration with the inclusion of these distant predictors.

However, as our updated reconstruction includes an additional pool of 12 predictors, and Reviewer 2, comment R2.6. suggested that the inclusion of this section is not necessary, we have decided to omit this section.

As our reconstruction procedure uses PCA regression, the predictions of discharge are based on common shared variance between the tree-ring predictors. The PC1 and PC2 loadings in the

Supplementary Table 1 contain information on how each tree-ring series relates to these two ‘shared variance’ series. Additionally, based on our validation results in Fig. 1a (with Editor Comment E1) we find that the ‘nests’ that include more predictors typically have higher skill than the nests with a lower number of predictors.

RI.7 Fig S7. The authors said " The comparison illustrates that the multi-decadal dry and wet periods over the basin suggested by our JAS reconstruction are also suggested by larger scale reconstructions of spatial drought variability." It's not clear, the corr coeff. is very low $r=0.27$, what can be really learned from this comparison? A low-pass filtering form may bring more information.

We have updated our Supplementary Fig. S7 below (Fig. 8).

Fig. 8. A comparison between standardized versions of our JAS discharge reconstruction at Bahadurabad (in red) and a spatial average of the June-July-August (JJA) mean Palmer Drought Severity Index (PDSI) over the Brahmaputra watershed reconstructed by the Monsoon Asia Drought Atlas (MADA - ²⁵) between 1309-2004 C.E. (Pearson $r=0.27$, $n=696$, $p<0.001$). While both datasets share many of the underlying predictors, they have different reconstruction target fields (Brahmaputra discharge vs gridded PDSI) and were produced using different reconstruction methods (Bayesian Regression vs spatial point-by-point regression). The lower panel compares 50-year low-pass filtered versions of both reconstruction highlighting that multi-decadal dry and wet periods over the basin suggested by our JAS reconstruction are also suggested by larger scale reconstructions of spatial drought variability. The 2 low-pass filtered series correlate at 0.52. While the low-pass filtered versions of the reconstructions show good visual correspondence, we note that this correlation is not ‘statistically significant’ at $p<0.05$ using a 2-tailed t-test considering the small effective sample size of 13.9 years (calculated as $696/50$). The correlation needed to obtain a p value < 0.05 for a sample size of 13.9 is 0.5342.

RI.8 *On which basis has the period 1417-2004 been divided into sub-periods of 10 years? Why 10 years? It would be helpful to know how these findings will be change/robust into others time size. Did the authors check the stationarity of the discharge reconstruction? Do the last 50 years of the discharge reconstruction follow the same probability distribution of the 600 years?*

We agree that the choice of 10-year periods was relatively arbitrary and have taken out this section. As in the initial submission, we now just compare the instrumental period mean against the reconstruction.

Yes, over the instrumental calibration-validation period the reconstruction matches the instrumental data (mean, variance, and density profile) very well (reply to comment R1.2, Fig. 3). The kernel density profiles in Fig. 3 also suggest that the instrumental period is not representative of the range of natural variability in the system. The differences of means test between the means during these periods show is significant.

RI.9 *Line 354: The authors estimate the "recurrence interval of flood events" using 1000 bootstrap values, i.e random sampling with replacement from 30 years of each dataset (instrumental observations, instrumental reconstructions, full reconstruction and CMIP5). Please detail more clearly the procedure used.*

We added some more text to the Methods section to clarify this better.

Updated Lines 421-431: *"To calculate the recurrence interval of flood events we computed 1,000 bootstrapped draws with replacement of 30 years each from (i) the instrumental observations (1956-2011 C.E.), (ii) the reconstruction over the instrumental period (1956-2004 C.E.), (iii) the full reconstruction period (1309-2004 C.E.), (iv) CMIP5 RCP8.5 2050-2074 C.E. period runoff simulations, and (v) CMIP5 RCP8.5 2075-2099 C.E. period runoff simulations. For the draws from the reconstruction we used the median reconstruction, while for the CMIP5 data we included all 42 ensemble members of the 20-model suite to represent the full range of variability in the model simulations of discharge. In each draw, we calculated the percentile (P) of the 2007 C.E. JAS discharge of 48,800m³/s. We then calculated the return interval as 100/(100-P), to give an estimate of the likelihood of the occurrence of high discharge related flood hazard in any given year. For example if in a random draw of 30 years, the 2007 C.E. flood year discharge placed in the 90th percentile, its return interval probability would be once every 100/(100-90)=10 years. However, if it placed in 80th percentile its return interval would be more frequent at once every 100/(100-80)=5 years."*

RI.10 *The absence of relationship between the monsoon seasonal flow and the modes of climate variability as ENSO and IOD looks surprising and should be detailed and discussed more: what analysis was performed to claim this finding?*

Please see Fig. 9 below that is now included as Supplementary Fig. S10

Fig. 9. Spatial correlation between mean JAS CRU Ts 4.01 precipitation and (a) mean JAS Oceanic Niño Index (ONI) and (b) mean JAS Indian Ocean Dipole (IOD) conditions based on the Dipole Mode Index (DMI) between 1950-2016. Over South Asia, correlations between JAS precipitation and ONI are the strongest over western India and Pakistan and are largely located outside the Brahmaputra basin. Correlations between DMI and regional precipitation are not significant. Only correlations significant at $p < 0.05$ using a 2-tailed t-test are shown in (a) and (b). The two lower panel plots show standardised anomalies of mean JAS Brahmaputra discharge at Bahadurabad plotted against standardised anomalies (c) JAS ONI and (d) JAS DMI. Neither correlation described in c. and d. is significant at $p < 0.05$ using a 2-tailed t-test. These analyses suggest that neither the El Niño Southern Oscillation (ENSO) nor the IOD drive Brahmaputra basin precipitation and streamflow, though we note that the relationship in c. may be non-stationary. ONI and DMI indices are multiplied by -1. ONI data is available here: <https://catalog.data.gov/dataset/climate-prediction-center-cpc-oceanic-nino-index>, and DMI data at this link: https://psl.noaa.gov/gcos_wgsp/Timeseries/DMI/.

RI.11 line 72-73 : There is no information about the Bahadurabad discharge record (provider, availability, precision, quality, sample,...)

In the Data Availability section we mention that the Bahadurabad discharge data is from the Bangladesh Meteorological Department. We also include the discharge data as Supplementary Data S1.

RI.13 line 131-132 : There is no information about the regional precipitation dataset used. How and why were TRMM, GPCC and CRU selected and used?

CRU and GPCC are among the only long-term gridded datasets that cover South Asia. The Aphrodite product ended in 2011²⁶. However, for a base precipitation climatology of the region shown in Supplementary Fig. S1, we chose TRMM as it has a higher spatial resolution and is merged satellite-gauge product²⁷, that overcomes some of the limitations of low station coverage in most of South Asia.

RI.14 Fig1: please give the signification of correlation coefficients

We modified figure by masking all insignificant correlations.

RI.15 Line 143 : correct "that that".

Done.

.....
Reviewer #2 (Remarks to the Author):

I have completed my review of the manuscript (NCOMMS-20-04754-T) "Six centuries of reconstructed Brahmaputra River flow demonstrate underestimated flood risk" by Rao et al. I found the manuscript to be fairly straightforward and well written. This is interesting and important research for our discipline, and I have no doubt that this manuscript will be a useful addition to the literature. I offer discussion of a couple of issues for consideration.

There are essentially two main arguments in this manuscript; 1) that the tree-ring records are faithful proxies of monsoon season river discharge and 2) that the estimates of seasonal discharge derived from the tree-ring records provide reliable estimates of "flood risk".

I believe that the analyses described in the manuscript make it clear that the discharge reconstruction is statistically sound overall and does a good job of estimating the mean discharge. And certainly seasonal discharge is related to flood risk. However, in my opinion the manuscript somewhat overstates the usefulness of the reconstruction to provide reliable estimation of changes in "flood risk". From my perspective this is really the only serious potential issue with this manuscript so I will discuss my overall thoughts on this matter first followed by some specific questions and suggestions and then a few relatively minor items (many of these can also be found on an attached "track changes" version of the submitted manuscript).

Thank you for the track changes document. We have incorporated these changes as best we could, though we might have missed some as the version we were able to work off from was a pdf file.

On line 103 the authors describe how seasonal discharge and maximum 10-day mean discharge (essentially a proxy of major flooding) are coupled, which is not surprising, but (here I disagree with the authors' perspective) the relationship is not very strong (less than half the variance is explained). The essential argument of the manuscript is then given in the Conclusions (line 254)

“We found the magnitude of peak discharge during floods in the JAS monsoon season to be tightly coupled to mean discharge for the entire JAS monsoon season. This allows us to use our tree ring based reconstruction to inform our understanding about both past and long-term variability in flood risk in the watershed.”

To rephrase it less elegantly- because mean seasonal discharge is “coupled” with maximum 10-day mean discharge, and the tree-ring proxies can explain 53% of the variance in seasonal discharge, the reconstruction is a good proxy of “flood risk”.

So my questions about this argument are

R2.1 *If estimating “flood risk” is the goal, why not reconstruct maximum 10-day discharge? It seems like it would be a lot less convoluted.*

While the tree-ring records that we use are sensitive to regional hydroclimate, given the characteristics of the hydrology in the region with the long monsoon season, tree-growth is more related to seasonal soil moisture than to 10-day (or short-term) precipitation. While the short-term 10-day discharge is positively related to seasonal precipitation, as we show in Fig. 2b, we are more confident in using the trees to develop a seasonal reconstruction of discharge as opposed to 10-day annual maximum discharge. Though we definitely agree that reconstructing these short-term extremes directly (e.g. as in refs ^{28,29}) would have been preferable. We have tried to better clarify that we reconstruct discharge and not flood events throughout the manuscript, while noting that these high discharge and flood hazard are related. Also see reply to comment R2.2 below.

R2.2 *Figure 3a, and especially 3b indicates that there are many years when the reconstructed discharge exceeds the value (48.8k m³) indicating a “flood”, but no flood is recorded and there are years when there are documented floods but reconstructed discharge values are relatively low. I am curious to know what the actual number of “hits/misses” are. The authors deal with this to an extent in their SEA discussion, which I think was well done. However the SEA approaches the issue from a mean value perspective, which I think tends to blur the actual efficacy of the model. In any case I think it is important that the authors at least touch on the fact that there are many instances where the model falsely indicates what the authors have defined as “floods”, or fails to indicate a recorded flood.*

We have tried to better phrase our language in our manuscript regarding ‘flood risk’ and ‘flood hazard’ (as suggested by the Editor and Reviewer 3). We also clarify that we use the tree-rings to reconstruct discharge, and that while floods have almost always co-occurred in years with high discharge, high discharge by itself does not equal flooding. In other words, P(high discharge|flood) is much greater than P(flood|high discharge).

Updated lines 270-283: *“In the instrumental observations, mean JAS discharge exceeded 48,800 m³/s in 13 years (Fig. 2b). Despite high discharge during these 13 years, more than half of these years (n=7) experienced no flood. While our recurrence interval analysis focusses on the frequency of high discharge that is associated with the likelihood of flood hazard, many other factors play a role in determining whether high discharge translates to a flood event. These may include rainfall intensity and pattern, landscape heterogeneities, antecedent soil moisture conditions, and land use and forest cover change^{11,12,30}. Our return interval analyses also rely on the assumption that these high discharges will continue to be associated with an increased likelihood of flood hazard in the future, disregarding (for example) potential changes in policy,*

land use, or infrastructure that may ameliorate 'flood risk'. The occurrence of a flood event that impacts society is however closely intertwined with highly localised human exposure and vulnerability^{16,17}. Therefore, our calculations of underestimated high discharge and associated likelihood of flood hazard in the return interval analyses in Fig. 3b only contributes one component of the multiple dimensions of flood risk."

Updated lines 312-313: "A limitation of our analyses regarding flood hazard is that we reconstruct Brahmaputra mean JAS monsoon season discharge and not flood years per se"

Updated lines 318-319: "*Additionally, we focus on the likelihood of high discharge as a proxy for flood hazard, and not on flood exposure and vulnerability^{15,18}.*"

2.3 Another argument made by the authors is that the instrumental period (1956-2016) (and especially the early portion 1956-1989) has been unusually dry compared to the full reconstruction (e.g. see line 150). This argument is primarily supported by a return interval analyses (RIA) that shows that the RI of "floods" is nearly twice as long in the instrumental period as compared to the overall reconstruction. Looking at Fig. 3a one can see that there are two pronounced low-flow periods between 2004-2016 so I think this argument would be strengthened if readers are provided the value for the 1956-2004 instrumental mean. Since this is the period in common with the reconstruction, I believe readers would like to know how different it might be from the 1956-2016 mean. Also we are told in the Methods that the RI was performed on the 1956-2004 period but in the Results (line 211) the 1956-2016 period is indicated. So it is not clear whether the period used for the RI is problematic.

We corrected the error in our Methods section in **Line 422** to specify that the return interval analysis was done using the full instrumental period (1956-2011, see reply to comment R1.3). However, we now include a return interval analysis for the 1956-2004 and 1956-1998 periods in Supplementary Fig. S9 (see Fig. 10 below). We found that the median return interval for discharge in excess of 48,800 m³/s across all three time periods remains 4.35 with small differences in the interquartile range. Though we acknowledge that this analysis is limited by only having 56 years of data to resample from.

Additionally, based on this recommendation we have also tried to better clarify the time period used in all the analyses presented in our manuscript.

Fig. 10 Recurrence interval (in years) of discharge greater than the 2007 flood year in three different time periods of the observed instrumental data, i. 1956-2011, ii. 1956-2004, and iii. 1956-1998. The first period includes all instrumental observations, the second is the period of overlap between the instrumental observations and the reconstruction, and the third is the calibration-validation period for the reconstruction. The median recurrence interval is 4.35 for all three time periods, though there are slight differences in the range of variability across the 1,000 draws of 30-years with replacement. We note that the lack of difference in the median return interval could be in part due the short instrumental series.

I think this is really the only overall issue with the paper. I think as a streamflow reconstruction, there are really no problems, but as a proxy for flood risk I don't think it is as persuasive as it could be.

Following are some specific notes. Other more minor items can be found on the track changes doc.

2.4 Lines 103-107 “closely coupled” and “strongly associated” might be overstating the relationship. We’re talking about less than 50% of the variance. Also I think it would be helpful to state whether the discharge data described here and in Fig 2b. are the instrumental or reconstructed. I assume instrumental since values after 2004 are given but it would be help to be specific. It could be useful to also show the relationship with reconstructed discharge.

We now clarify that this entire section is about the observational record using the sub-heading: “Seasonal hydrograph and recent flood events in the observational record”.

Updated Line 98-99: “As tree rings typically provide information regarding seasonal hydroclimate³¹, we then attempted to determine whether JAS monsoon season discharge is closely coupled **related to** sub-seasonal flow at the 10-day timescale relevant to regional flooding.”

We removed the next line that used the phrase ‘strongly associated’, in addition to being more careful about our choice of adjectives throughout the paper.

2.5 Lines 108-117 I’m not sure I understand how this discussion advances the argument that the recon is a good proxy for flooding. I assume the point is that flood years usually co-occur with lengthy periods of above average flow. However examination of Fig.S3 appears to show that it is very common for above average flows of long duration to occur in years when no flood was recorded.

See reply to comment R2.2 that we clarify that high discharge doesn't mean flooding, but as the reviewer said, the two usually co-occur.

2.6 Lines 135-147 I don't find this necessary, but I suppose someone might question the more distant chrons being included. I think this could easily be moved to the Suppl Mat.
We took this section out.

2.7 Lines 150-152 I am curious why the full instrumental period mean 1956-2016 is used when the recon only extends to 2004? Maybe it doesn't make any difference, but since the case is being made that the instrumental period is lower than the recon I think readers would want to know how the means compare over the common period. I think this is especially relevant since the 1956-2004 period is used in the other analyses.

See reply to reviewer comment R1.2 where we include text in the manuscript that describes that choice of 1956-2011, 1956-2004, or 1956-1998 to calculate the instrumental mean did not affect our main finding. Our monthly mean dataset is now truncated at 2011 due to many missing days in the daily data between 2012 and 2016.

2.8 Line 211 The authors state "we used the instrumental data (1956-2016),..." Is this correct? It says 1956-2004 in Methods

We corrected it to 1956-2011 in the methods but also provide results for different time periods 1956-2004 and 1956-1998. Also see reply to comment R2.3.

2.9 Line 401 Using flood data as event years in an SEA of discharge seems problematic given that they are not entirely independent, however, I think the authors have done a really good job of estimating the true significance of the difference in means of the flood years. I am curious however whether using the values (upper) for the confidence intervals for the draws composing the 'pseudo-flood years', would still result in a significant difference.

This is good point, especially as the relationship between high discharge and flood years is much weaker in the pre-instrumental period. We repeated our SEA using the 12 pre-instrumental flood years. While there is a high discharge 'signal' in these flood years it is much weaker than over the instrumental period (Fig. 11). We include this figure as Supplementary Fig. S7. We also added text in the manuscript highlighting how this relationship is much weaker and not 'statistically significant' for flood years prior to 1956.

Also see reviewer comment R1.5 for modification to the manuscript text.

Fig. 11 Superposed Epoch Analysis (SEA) for discharge in 12 historical flood years that occurred prior to the start of instrumental observations in 1956. The vertical lines on the response bars are the 5th, 50th, and 95th percentiles of mean flow across 495 unique draws of 8 flood years at random out of 12. The horizontal dotted lines indicate the threshold required for epochal anomalies to be statistically significant using random bootstrapping at three different statistical thresholds. These thresholds were calculated by compositing 10,000 draws of 8 years at random (or ‘pseudo-flood years’) from the reconstruction between 1780 and 2004. The relationship between high discharge during flood years is much weaker than that for just the instrumental period flood years (Main Text, Fig. 2b) and for all 16 flood years (Main Text, Fig. 4a). The median response of the 495 unique draws of 8 flood years out of 12 is not significant at $p < 0.05$ when compared to 10,000 draws of 8 ‘pseudo-flood’ years at random.

2.10 Line 289 Data Availability- I think that each of the 16 tree-rings sites used in the analyses needs to be cited in the Refs. or Supp. Mat refs. Also I think that if reproducibility is important the supp mat or the ITRDB archived materials need to be much more specific. For example, not only the full reconstruction but each model run as well as the calibration and verification stats for each model, the sample size (by year) for each model etc. should be in the SM or archived at the ITRDB. We better cite each individual tree-ring series used. We also include our R code used to develop our reconstruction along with all standardised tree-ring chronologies and their metadata in the Supplementary Data. This along with the included discharge data should allow any users to reproduce our reconstruction results here.

We include the calibration-validation statistics for each ‘nest’ including their sample size (i.e. number of chronologies used) in Supplementary Fig. S5. The same figure is presented here as Fig. 1A (with Editor comment E1).

2.11 Figure 3a The recon seems to do a poor job on the extremes and especially the dry extremes (which seems unusual). It might be useful to apply the bias correction technique discussed by Robeson et al. (2019).

We agree, based on our experience with tree-ring reconstructions we found this to be surprising as well.

We actually did try to implement Robeson et al.'s bias correction technique prior to our first submission. We tried to repeat it again with our updated reconstruction as well. We found that it did not improve model skill (it was reduced in many iterations).

Our sense is that Q-Q mapping needs a long enough observational dataset to accurately linearly regress all (100) percentiles of the estimated series to the target observational series. However, as we are working with such a short observational series, unlike that available for the Colorado River, this method did not improve the overall model statistics. Additionally, the instrumental data we use is likely not representative of the range of natural variability in the system (it is much drier). As Q-Q mapping 'adjusts' the reconstruction such that it is informed by the maximum and minimum values of the 'observed pdf', the procedure imposes an artificial bound on the range of pre-instrumental variability allowed in the reconstruction. Models with this kind of 'extrapolation' where the range of predicted estimates are greater or less than observed values are not recommended for 'bias correction'³². Robeson et al. acknowledge this limitation in their paper as well.

2.12 Figure S8. Table S3. There are instances where the term "cyclone" or "cyclonic" are used but it is not specifically noted as a "tropical cyclone", which I think would be better.

In our original submission we felt it was important to include this as the more common association with flooding in Bangladesh is with coastal flooding. However, we took this section out in the updated manuscript.

2.13 Table S1. Is there any order to these sites? It might be helpful to list them by Lat and Long so that readers can get an idea of where individual sites are on the map.

These sites are ordered based on chronology length. We now include a note about this in Supplementary Table S1. As we include all series used and their metadata (including latitude and longitudes) in the supplementary, that should help users locate and use any of the data if they wish to.

Please let me know if I may offer any clarification or be of any other assistance.

Sincerely,
Matthew Therrell
Professor
Refs

Robeson, S. M., Maxwell, J. T., & Ficklin, D. L. (2020). Bias correction of paleoclimatic reconstructions: A new look at 1,200+ years of Upper Colorado

.....
Reviewer #3 (Remarks to the Author):

The authors presented a study reconstructing the flow of the Brahmaputra river based on tree-rings over the last six centuries. The study conclude that "flood risk is underestimated by the instrumental record", and that "recent observations to will be insufficient to determine the degree of flood risk for the coming decades". The importance of such studies supporting this is high, mostly due to the large impact that floods have on millions of people living in the floodplains of

this river, but also to support one more time, that systematic records are not enough to determine properly flood hazard.

Yet the manuscript provided by Rao et al., fails under my opinion is the following items, which I consider major problems, specially point 1 and 2, preventing me, unfortunately, to accept this ms:

R3.1 Authors are mixing concepts between flood hazard and flood risk along the entire ms. This imply that the ms is not technically sound, since not vulnerability, neighed exposure assessment is provided. Similarly, there are some other concept in the ms used that are somehow not commonly used in hydrology as, climatological flow (although this is less importance)

Authors present “flood risk” from now I consider it as “flood hazard” without analyzing stationarity and providing a way to compare the results with classical flood hazard assessment, which normally use the concept of return period. The stationarity, highly discussed in hydrology and paleohydrology, is not mentioned neither.

Thank you for your comments. Our manuscript now focusses on flood hazard as opposed to flood risk (also see response to editor comment E2). In our manuscript we reconstruct discharge and show the instrumental data doesn't capture the full range of ‘natural variability’ in the system. Thus, even if the discharge process is ‘stationary’ we show it could be stochastic with memory (short or long) and may be quasi-periodic. Hence, we see persistent regimes even if these regimes may themselves be generated randomly or be externally forced by teleconnection patterns, volcanic eruptions, the ‘global monsoon’ etc. Notwithstanding, we use the tree-ring data as a proxy to have insight regarding these regimes. We also find that this variability is not observed in out short instrumental records. We note that these findings are based on the assumption that the regression fit of the proxy data in the recent period holds for all past time. However, if the tree ring PCs exhibit very different statistics in the past then we get a different distribution for the previous 600 years just from that by applying the ‘stationary’ regression. We observe evidence of this, as we find the instrumental period is drier than the recommendation. Also see reply to reviewer comment R1.1 where we describe common shared variability between Brahmaputra River variability and predictive information in the tree ring predictor pool.

We also explicitly acknowledge the limitation of our assumption that high discharge is related to flood hazard as follows:

Lines 270-283: *“In the instrumental observations, mean JAS discharge exceeded 48,800 m³/s in 13 years (Fig. 2b). Despite high discharge during these 13 years, more than half of these years (n=7) experienced no flood. While our recurrence interval analysis focusses on the frequency of high discharge that is associated with the likelihood of flood hazard, many other factors play a role in determining whether high discharge translates to a flood event. These may include rainfall intensity and pattern, landscape heterogeneities, antecedent soil moisture conditions, and land use and forest cover change^{11,12,30}. Our return interval analyses also rely on the assumption that these high discharges will continue to be associated with an increased likelihood of flood hazard in the future, disregarding (for example) potential changes in policy, land use, or infrastructure that may ameliorate ‘flood risk’. The occurrence of a flood event that impacts society is however closely intertwined with highly localised human exposure and vulnerability^{16,17}. Therefore, our calculations of underestimated high discharge and associated likelihood of flood hazard in the*

return interval analyses in Fig. 3b only contributes one component of the multiple dimensions of flood risk.”

The bootstrapping approach we use here for our return interval analysis resamples only available data without making assumptions of the underlying distribution or process. We feel that this approach makes it intuitive and easy to understand. We also provide additional text to explain our method better (see reply to reviewer comment R1.9).

R3.2 The outcomes are based on a relationship described by the 53% explained variance. I am not sure if we can call this as strong evidence.

Please see response to editor comment E1 and reviewer comment R1.1.

Results could be novel, but the general conclusion about the drawbacks of using short records is not.

We use this more as a motivation of why this kind of research is useful (also see response to reviewer comment R1.0).

R3.3 I am missing key references to palaeohydrology in the region, as well as their inclusion in the discussion related to wet/dry periods.

We have included some additional references to paleohydrology in Asia and in the watershed including Cook et al. 2003³³, Gou et al. 2007¹, Liu et al. 2012³⁴, Y. Chen et al 2020⁷, Li et al. 2017³⁵, He et al. 2018⁸. Yang et al. 2014⁴, Chowdhury et al. 2016³⁶, Thapa et al. 2017³⁷, Islam et al. 2018³⁸, Maxwell et al. 2018³⁹, Shi et al. 2018⁴⁰, Li et al. 2019⁶, Wang et al. 2020⁹.

Also see comment R3.39 below for comments on wet and dry periods.

R3.4 The methods here used are broadly accepted by the tree-ring community, but an identification of the assumption, and including a discussion of the limitations of the results is missing in the ms. Thus, I should accept that reconstructing with the 53% of explained variance is considered ok, as I did not see any discussion related to potential drawbacks or limitations. An interval confidence level should be also provided in the text, and not only graphically masked in the Figure 3 (gray bars).

The full confidence interval of our calibration-validation statistics is provided in our manuscript (and here as Fig. 1). Almost all of our analyses are based on bootstrapping and Bayesian methods and therefore explicitly incorporate uncertainty. We also now include the confidence interval around the mean in **Lines 148-157**.

“The reconstructed discharge estimates during the observational period are shown in Fig. 3a and our full reconstruction between 1309-2004 C.E. is presented in Fig. 3b. While the reconstruction is calibrated to the instrumental mean and variance in the reconstruction procedure (Fig. 3 c-d), we found that the mean reconstructed discharge over the full reconstructed period between 1309-2004 C.E. was significantly higher than the instrumental mean between 1956-2011 C.E. ($46,993 \pm 812 \text{ m}^3/\text{s}$ cf. $43,350 \text{ m}^3/\text{s}$, difference of means = $3,644 \text{ m}^3/\text{s}$, t -statistic = 5.11, $p < 0.01$) (Fig. 3a). The uncertainty range around the mean was derived from the 5th and 95th percentiles of the means across all 400 iterations of the median reconstruction.”

R3.5 Only a RCP is used, without explain why this has been chosen. I imagine they are interested in only one extreme of expected impacts...

We were unable to put together a similar suite of RCP4.5 or RCP6.0 simulations with enough model and ensemble members to be confident in our projections of future discharge under these scenarios. The reason we wanted to only use models that have the same ensemble members between the ‘historical’ and ‘future’ runs is to be able to append the two runs together and compare the two.

As a better estimate of how sensitive our future discharge projections are to changes in expected warming in global mean annual temperatures, we split the 2050-2099 period into two halves as 2050-2074 and 2075-2099. We also calculated the expected changes in global mean annual temperature for this period relative to pre-industrial times between 1850-1880 (Fig. 12). This figure is now included as Supplementary Fig. S8.

Updated lines 234-241: “We divided the end-of-the-century CMIP5 RCP8.5 discharge simulations into two halves (2050-2074 C.E. and 2075-2099 C.E.) to estimate the sensitivity of our results to different levels of global mean warming relative to the pre-industrial era with continued anthropogenic carbon-dioxide emissions under the RCP8.5 scenario (+3.05°C by 2050-2074 C.E., and +4.30°C by 2075-2099 C.E.; Supplementary Fig. S8). The 3.05°C warming of global mean annual temperatures that we estimate here by 2050-2074 C.E. under RCP8.5 is roughly equivalent to the projected warming that will be achieved by 2099 C.E. under the lower emission RCP4.5 scenario (see Fig. 1 in ref. ⁴¹).”

Fig. 12 Expected change in global mean annual surface temperature between 2050-2074 and 2057-2099 relative to pre-industrial 1850-1880 conditions using CMIP-5 RCP8.5 projections. The multi-model median warming for these two periods is projected to be 3.05° and 4.30°C respectively. We used the same suite of 20 models and 42 ensemble members as in our modelled runoff calculations for this analysis. The full list of models and the respective ensemble members can be found in Supplementary Table, S1.

Other comments are below:

R3.6 The title is not technically sound. Authors talk about flood risk, but not risk assessment is provide. It seems they are unaware about the risk concept (as broadly defined, last GAR 2019), which must include an assessment of the vulnerability, exposure and hazard. Here authors don't show any assessment on exposure and vulnerability, and partially on hazard.

Our updated title is, 'Seven centuries of reconstructed Brahmaputra River discharge demonstrate underestimated high discharge and flood hazard frequency'. Also see reply to Editor Comment E2.

R3.7 Line 18: vulnerable? You mean exposed? Vulnerable includes how people is affected by any impact, in relation to its cultural or socio-economic characteristics (among others). And I guess is defined to people (or assets) living (being) in floodplains rather than floodplains itself, to which I guess it is better to use "recurrently affected by"

Updated line 18: *"The lower Brahmaputra River often floods during the monsoon season (July-August-September; JAS)."*

R3.8 Line 20-23: This sentence should be split in two, one talking on intensified summer monsoon and other in flood risk. In fact, the intensified monsoon is not the only component leading an increase on flood risk.

Updated lines 19-21: *"While most climate models predict an intensified monsoon through the twenty-first century, robust baseline estimates of natural climate variability and flood hazard for the region are limited by short and fragmentary observational records."*

We also mention that our climate model projections of runoff and our reconstruction only contribute one component of flood risk. See updated text in reply to reviewer comment R3.1.

R3.9 Line 24: Brahmaputra river

Corrected

R3.10 Line 24: mean JAS flow discharge.

We phrased it as "mean JAS Brahmaputra discharge".

R3.11 Lines 26-27: these findings are not novel, but expected, true?

That is correct. In the paper we cite Immerzeel, 2008⁴² who shows this is relationship in the Brahmaputra River and Milly et al. 2002¹³ who show that this is common in 'large' river basin in the world.

R3.12 Line 27-29: how these findings related with the existing paleofloods records?

This line is no longer in the abstract.

R3.13 Line 30: you are not providing flood risk. This is not technically sounds. The term "flood risk" is not well used in the entire manuscript.

Modified as: *"Further, flood hazard as determined by the recurrence frequency of high discharge"*

R3.14 Line 33-35: the philosophy here expressed “a focus on recent observations to evaluate flooding in the basin will be insufficient to determine the degree of flood risk” is not novel. In fact, statisticians and paleohydrologist have broadly stated the same idea since 1960’s. From the sentence, it seems that you ending with a general conclusion. I’d suggest here to highlight your own conclusions otherwise, it looks not novelty for me.

This observation is certainly a common theme in paleohydrology and paleoclimatology. However, this line immediately follows the abstract text that shows the implications of this in terms of the percentage underestimation of the likelihood of high discharge in the Brahmaputra River using instrumental observations alone.

R3.15 Line 45-50: May be it is a good idea to short this sentence or to list the items for a better read.

We have shortened this sentence as follows: “*These benefits include fish (a primary source of protein in the region), water to irrigate many seasonal rice varieties that need annual flood waters to survive, the deposition of fresh sediment to sustain the large inhabited riverine islands (known as chars), and the prevention of salt-water intrusion from the Bay of Bengal into the low-lying Sundarban delta*⁴³⁻⁴⁵.”

R3.16 Line 54-56: it sounds repetitive. The idea is already introduced in the lines 41-42.

We felt that it was too early to present results of Fig. 1a and Supplementary Figs. S2 and S3 until we mention why understanding monsoon discharge is important in lines 49-50.

R3.17 Line 69: I have read this paper and I did not found where is indicated that “flood risk” will increase in Brahmaputra. This is for me a major problem of the ms so far, I think you are missing concepts between flood hazard and flood risk. May be you assume that an increase in flood hazard results in an increase in flood risk, but technically is not sound.

We added a few additional citations here^{14,19,46} and have modified this line as follows:

“*This intensification of the monsoon, along with accelerated warming-driven glacial melt, is expected to cause greater flow in the Brahmaputra River*^{47,48} *and likelihood of flood hazard in the region*^{14,19,46,49}.”

R3.18 Line 79-76: I highly miss the existing bibliography on paleohydrology and historical hydrology here. For instance, Puni and Ravishanker (1983), contextualizing extreme flood event into long-term historical records. Kale et al., (1987), who suggested an increase on floods during last decades...or Kale et al., (1997, 2000) suggesting clustering of large flood events in the recent decades. In other part of India there are other works, like Ely et al., (1996) or Thomas et al., (2007). I guess some of these works, especially in the Easter part of India should be mentioned? How these findings relates to yours?

These studies that include geomorphic evidence such as slackwater flood deposits and field stratigraphy are either on much longer timescales or unfortunately do not cover the Brahmaputra River basin (Kale et al 1994⁵⁰, Ely et al. 1996⁵¹, Kale et al. 1996⁵², Thomas et al. 2007⁵³). We certainly acknowledge that such work would certainly help develop better and longer records of paleo-floods in the region. We mention the same in our ‘Conclusions’ and cite these studies.

R3.19 Line 77: using tree-rings as a surrogate of hydroclimate is a way to indirectly reconstruct “streamflow”. I miss here references that use tree-rings records to date and reconstruct individual past flood events. Some of them are: Balesteros-Canovas et al., 2015 or Wilhelm et al., 2019. The dendrogeomorphic methods used by Ballesteros-Cánovas et al., 2015 and Wilhelm et al. 2019⁵⁴ cannot be directly used in our study as their studies use flood-scar records to reconstruct individual flood events. However, this does represent a limitation of our study as we reconstruct discharge and not flood events.

We acknowledge this in **lines 311-318** as follows - *“A limitation of our analyses regarding flood hazard is that we reconstruct Brahmaputra mean JAS monsoon season discharge and not flood years per se. Improved paleo-hydrology work in other archives, such as geomorphic evidence and field stratigraphy⁵⁰⁻⁵³, the documentation of tree-ring flood-scars that can precisely date past flood events⁵⁴⁻⁵⁶, and additional tree-ring sampling in the region of traditional³⁷ and non-traditional species^{36,38,39} can help establish more skillful reconstructions of Brahmaputra discharge, its flooding history, and its flooding frequency in future work⁵⁷.”*

R3.20 Line 88: you mean between monsoon and seasonal flow discharge.

We modified **lines 82-86** as follows: *“We use these models and our reconstruction to evaluate two situations. The first is how the recurrence of high discharge events (used here as a proxy for flood hazard) in recent decades compares to longer-term estimates over the last several centuries. The second is how the increase in Brahmaputra River discharge caused by projected regional wetting⁴⁸ compares to natural climate variability estimated by the instrumental data and our tree-ring derived reconstruction.”*

R3.21 Line 85: why you use only RCP 8.5?

Please see reply to reviewer comment R3.5

R3.22 Line 86: I strongly disagree on the point: how flood risk in recent decades compares to longer-term estimates over the last several centuries. For the points indicated about the concept “risk”.

Please see reply to reviewer comment R3.20.

R3.23 Line 94: this information is mentioned already. Please don't repeat information.

We retained **line 94** as it presents additional information that annual discharge peaks between July-September. We use that to set up the next line that presents the annual hydrograph of discharge.

R3.24 Line 98-100: This sounds as a strong assumption. Do you have information supporting this? I don't have doubts about the relationship between tree-ring widths and temperature or even precipitation, although this provides a more complex figure. Translate this relation into streamflow would be ok, but I consider as an indirect relationship, the problem I see here is how to link exactly tree-ring width with maximum flood during 10-days. Do you have like a kind of model-based results to support how seasonal precipitation is transformed into a hydrography signal at the studied reach river? Or other kind of information. Otherwise, for me it is an assumption and you should treat it (and presented) properly in the ms.

We clarify that we reconstruct seasonal discharge and not flooding or peak discharge, while showing that these components are related. We also include correlations between upper basin

model simulated runoff and gauged discharge at Bahadurabad (Fig.4 and reply to reviewer comment 1.3.).

R3.25 Line 102: did you probe other combination of days? I imagine your above-mentioned assumption is based on this relationship but, did you test others combination of mean-hydrographs values?

We found the relationship between shorter-term (1-day, 5-day, 10-day, 20-day, and 30-day) discharge to all be well correlated with mean JAS discharge (Fig. 13). We used 10-day discharge as opposed to a shorter average as it represents a more extreme flood period in the basin²¹.

R3.26 Line 103-104: this is obvious.

We felt that documenting this relationship is important to explain why a reconstruction of past discharge for the Brahmaputra River is valuable.

R3.27 Line 113: what is climatological flow? You mean ordinary flow? Bankfull discharge? I not sure if climatological flow is a broadly accepted term in hydrology.

We changed to ‘median’ flow so as to use a more precise definition based on years of record.

R3.28 Line 122: do you think that 42 years is enough to calibrate and validate the relationship that then is used to reconstruct six centuries?

We definitely agree it would have been much better to have a longer calibration-validation period. However, 1956-1998 was the best trade-off we had between the last year of most of the tree ring data we have in the region and the discharge data doesn’t go back further. Though our model does show reasonable validation statistics considering the limited data available. Our reconstruction method gets around some of the limitations of the short dataset by developing multiple iterations of the reconstruction calibrated to and validated against different leave-10-out subsets of the 43 years of instrumental data available.

R3.29 Line 128 -130: Actually, this has more sense, as tree growths are physically connected to the moisture in soil provided, mostly, by precipitation. I think this should be appear first and then admit that, as consequence, an assumption about tree-ring widths and seasonal streamflow can be done. Providing here the correlations could be interesting.

We provide the loading of each tree-ring series on the Principal Component (PC1) and PC2 timeseries in the Supplementary Table S1 (also see reply to reviewer comment R1.4).

R3.30 Line 131: I am aware about tree-ring climate relationship, and see that 53% is ok, but at the end, it means that the reconstructed flow discharge could be or could not be...This should be discussed.

Please see reply to reviewer comment R1.1.

R3.31 Line 147: provide interval confidence data!

We include a confidence interval around the mean as $46,993 \pm 812 \text{ m}^3/$ derived from the 5th and 95th percentiles of the means across all 400 iterations of the median reconstruction.

R3.32 Line 171: how reliable is your reconstr. in this time i.e. 1435? I miss information about the reliability (EPS) of the used tree-ring chronologies.

The minimum number of tree-ring predictors series used in our reconstruction is 10. Each predictor series consists of multiple trees. We only used sections of the individual tree-ring chronologies that had an $\text{EPS} > 0.85$. However, the EPS threshold is not that relevant to our reconstruction as we never use a single chronology by itself. We also provide the calibration-validation statistics for each individual next for the full reconstruction (also see reply to editor comment E1, Fig. 1A).

We now mention this in **Lines 366-369**: *“We then ‘standardized’^{58,59} each of annual raw ring-width series using the signal free (SF) method⁶⁰ to reduce the influence of non-climatic growth factors on tree growth and maximize the preservation of common median frequency at decadal to*

centennial timescales and truncated each chronology to the section with an Expressed Population Signal > 0.85^{61,62}.”

R3.33 Line 201: I can not understand how you estimate flood risk relative to observational period. I assume that you are simply wrong, and instead you should have used the term “hazard”. In this case, you should reconsider your text and correct it. But the repetitive use of the term “risk” along the ms, make me think that you are really mixing concepts, and this is more problematic. You cannot talk about risk, at least, you are assessing the vulnerability and exposure of the population living in the floodplains of Brahmaputra.

We have rephrased our manuscript to be about flood hazard.

R3.34 Moreover, flood hazard is commonly based (in hydrology) in the concept of return periods, to which a large discussion about stationary has been risen last decades. You can have a look in a recent recompilation paper where paleo hydrology is taken into account for flood hazard assessment, and where all these concepts are threatred (Wilhelm et al., 2019). How your data should be compared with other results using the concept of return period? Could you provide this information?

We have tried to clarify that in our study we calculate the return period of high discharge events and not flood events. Therefore, these cannot be compared directly, and we acknowledge this in our manuscript. See reply to comment R3.19.

R3.35 Why you used only RCP 8.5? and not the entire set, or at least the two most extreme, scenarios

Please see response to comment R3.5.

R3.36 Line 228-229: it is a bit strange to read here that tropical cyclones are major cause of floods. May be this can be merged in the information provided in line 41-42.

And R3.37 Line 233-235: already mentioned and it is not novelty.

Re 3.36 & 3.37: We took out this section regarding tropical cyclones.

R3.38 Line 240. Then, you used this dataset for your tree-ring / streamflow relationship?

Lines 231-232: Yes, we used the full instrumental discharge data available between 1956-2011.

R3.39 Line 227: I miss a discussion with the long-term climate-flood linkages provided by paleohydrologist in the region.

While we were not able to obtain the data for the reconstructions by Liu et al. 2012³⁴, He et al. 2018⁸, and Wang et al. 2020⁹ our results show good visual agreement in terms of wet and dry period found by these studies in the southern Tibetan Plateau. The southern Tibetan Plateau constitutes the upper part of the Brahmaputra basin. These studies are referenced in our paper in our section “Reconstruction of past discharge” to further support the dry and wet periods we find. As a visual comparison for the reviewer we include screenshots of three different hydroclimate reconstruction in Fig. 14. We also compare our reconstruction against the Monsoon Asia Drought Atlas²⁵ reconstructed PDSI for the region and show that the reconstructions agree well (Fig 8).

Fig. 14 Comparison between (a.) May-June self-calibrating Palmer Drought Severity Index (scPDSI) reconstruction by He et al. 2018 (ref - ⁸) between 1190-2010 C.E. for the south-central Tibetan Plateau, (b.) May-June sc-PDSI reconstruction for the Southeastern Tibetan Plateau between 1135-2010 C.E. by Wang et al. 2020 (ref. - ⁹) (c.) a prior June to current July precipitation reconstruction for the Southern Tibetan Plateau by Liu et al. 2012 (ref. - ³⁴) and (d.) our reconstruction of mean JAS Brahmaputra River discharge at the Bahadurabad gauge in Bangladesh between 1309-2004 C.E. Low-frequency variability is highlighted in (a.) using a 21-year Fast Fourier Transform (FFT), in (b.) using a 30-year low pass filter, in (c.) using a 11 year FFT and in (d.) using a 50-year low-pass filter. All three reconstructions show good correspondence in most wet and dry periods. Note that the four reconstructions are for different seasons, variables, regions and are largely independent in terms of tree-ring predictors used.

R3.40 Line 279: data availability. You should include the scripts you used, and specify which computational language you used.

We now include our reconstruction code and the datasets we use within Supplementary Data 1.

R3.41 Line 343: recurrence interval. This way to treat the flow data is different than the one normally accepted in hydrology. Why did you use this way, which advantages has in relation to the classical one, should be considered as a novel approach?

Bootstrapping is certainly not novel⁶³. We use bootstrapping as its results are not dependent on the choice of distribution fit or model parameters, which we feel makes the results presented relatively more intuitive and easy to understand.

References

- 1 Gou, X. *et al.* Streamflow variations of the Yellow River over the past 593 years in western China reconstructed from tree rings. *Water Resour. Res.* **43**, doi:10.1029/2006wr005705 (2007).
- 2 Buckley, B. M. *et al.* Climate as a contributing factor in the demise of Angkor, Cambodia. *Proceedings of the National Academy of Sciences* **107**, 6748-6752, doi:10.1073/pnas.0910827107 (2010).
- 3 Feng, S., Hu, Q., Wu, Q. & Mann, M. E. A Gridded Reconstruction of Warm Season Precipitation for Asia Spanning the Past Half Millennium. *J. Clim.* **26**, 2192-2204, doi:10.1175/jcli-d-12-00099.1 (2013).
- 4 Yang, B. *et al.* A 3,500-year tree-ring record of annual precipitation on the northeastern Tibetan Plateau. *Proceedings of the National Academy of Sciences* **111**, 2903-2908, doi:10.1073/pnas.1319238111 (2014).
- 5 Hessel, A. E. *et al.* Past and future drought in Mongolia. *Science Advances* **4**, e1701832, doi:10.1126/sciadv.1701832 (2018).
- 6 Li, J. *et al.* Deciphering Human Contributions to Yellow River Flow Reductions and Downstream Drying Using Centuries-Long Tree Ring Records. *Geophys. Res. Lett.* **46**, 898-905, doi:10.1029/2018gl081090 (2019).
- 7 Chen, Y. *et al.* Precipitation variations recorded in tree rings from the upper Salween and Brahmaputra River valleys, China. *Ecol. Indicators* **113**, 106189, doi:<https://doi.org/10.1016/j.ecolind.2020.106189> (2020).
- 8 He, M., Bräuning, A., Griesinger, J., Hochreuther, P. & Wernicke, J. May–June drought reconstruction over the past 821 years on the south-central Tibetan Plateau derived from tree-ring width series. *Dendrochronologia* **47**, 48-57, doi:<https://doi.org/10.1016/j.dendro.2017.12.006> (2018).
- 9 Wang, J., Yang, B. & Ljungqvist, F. C. Moisture and Temperature Covariability over the Southeastern Tibetan Plateau during the Past Nine Centuries. *J. Clim.* **0**, null, doi:10.1175/jcli-d-19-0363.1 (2020).
- 10 Ljungqvist, F. C. *et al.* Ranking of tree-ring based hydroclimate reconstructions of the past millennium. *Quat. Sci. Rev.* **230**, 106074, doi:<https://doi.org/10.1016/j.quascirev.2019.106074> (2020).
- 11 Cox, D. *et al.* Reinforcing flood-risk estimation. *Philosophical Transactions of the Royal Society of London. Series A: Mathematical, Physical and Engineering Sciences* **360**, 1373-1387, doi:doi:10.1098/rsta.2002.1005 (2002).

- 12 Bradshaw, C. J. A., Sodhi, N. S., Pem, K. S. H. & Brook, B. W. Global evidence that
deforestation amplifies flood risk and severity in the developing world. *Global*
Change Biol. **13**, 2379-2395, doi:10.1111/j.1365-2486.2007.01446.x (2007).
- 13 Milly, P. C. D., Wetherald, R. T., Dunne, K. A. & Delworth, T. L. Increasing risk of
great floods in a changing climate. *Nature* **415**, 514-517, doi:10.1038/415514a (2002).
- 14 Hirabayashi, Y. *et al.* Global flood risk under climate change. *Nature Climate Change*
3, 816-821, doi:10.1038/nclimate1911 (2013).
- 15 Kundzewicz, Z. W., Hegger, D. L. T., Matczak, P. & Driessen, P. P. J. Opinion: Flood-
risk reduction: Structural measures and diverse strategies. *Proceedings of the National*
Academy of Sciences **115**, 12321-12325, doi:10.1073/pnas.1818227115 (2018).
- 16 Winsemius, H. C., Van Beek, L. P. H., Jongman, B., Ward, P. J. & Bouwman, A. A
framework for global river flood risk assessments. *Hydrol. Earth Syst. Sci.* **17**,
18711892, doi:10.5194/hess-17-1871-2013 (2013).
- 17 McGlade, J. *et al.* (UN Office for Disaster Risk Reduction, 2019).
- 18 Winsemius, H. C. *et al.* Global drivers of future river flood risk. *Nature Climate*
Change **6**, 381-385, doi:10.1038/nclimate2893 (2016).
- 19 Gain, A. K., Immerzeel, W. W., Sperna Weiland, F. C. & Bierkens, M. F. P. Impact of
climate change on the stream flow of the lower Brahmaputra: trends in high and low
flows based on discharge-weighted ensemble modelling. *Hydrol. Earth Syst. Sci.* **15**,
1537-1545, doi:10.5194/hess-15-1537-2011 (2011).
- 20 Jian, J., Webster, P. J. & Hoyos, C. D. Large-scale controls on Ganges and Brahmaputra
river discharge on intraseasonal and seasonal time-scales. *Q. J. Roy. Meteorol. Soc.* **135**,
353-370, doi:10.1002/qj.384 (2009).
- 21 Webster, P. J. *et al.* Extended-Range Probabilistic Forecasts of Ganges and Brahmaputra
Floods in Bangladesh. *Bull. Am. Meteorol. Soc.* **91**, 1493-1514,
doi:10.1175/2010bams2911.1 (2010).
- 22 Grill, G. *et al.* Mapping the world's free-flowing rivers. *Nature* **569**, 215-
221, doi:10.1038/s41586-019-1111-9 (2019).
- 23 Harrigan, S. *et al.* GloFAS-ERA5 operational global river discharge reanalysis 1979-
present. *Earth Syst. Sci. Data Discuss.* **2020**, 1-23, doi:10.5194/essd-2019-232 (2020).
- 24 Huang, Q., Long, D., Du, M., Han, Z. & Han, P. Daily continuous river discharge
estimation for ungauged basins using a hydrologic model calibrated by satellite altimetry:
Implications for the SWOT mission. *Water Resour. Res.* **in press**, e2020WR027309,
doi:10.1029/2020wr027309 (2020).
- 25 Cook, E. R. *et al.* Asian Monsoon Failure and Megadrought During the Last Millennium.
Science **328**, 486-489, doi:10.1126/science.1185188 (2010).
- 26 Sun, Q. *et al.* A Review of Global Precipitation Data Sets: Data Sources, Estimation, and
Intercomparisons. *Rev. Geophys.* **56**, 79-107, doi:10.1002/2017rg000574 (2018).
- 27 Huffman, G. J. *et al.* The TRMM Multisatellite Precipitation Analysis (TMPA):
Quasi-Global, Multiyear, Combined-Sensor Precipitation Estimates at Fine Scales. *J.*
Hydrometeorol. **8**, 38-55, doi:10.1175/jhm560.1 (2007).
- 28 Howard, I. M. & Stahle, D. W. Tree-Ring Reconstruction of Single-Day Precipitation
Totals over Eastern Colorado. *Monthly Weather Review* **148**, 597-612,
doi:10.1175/mwr-d-19-0114.1 (2020).
- 29 Steinschneider, S., Ho, M., Williams, A. P., Cook, E. R. & Lall, U. A 500-Year Tree
Ring-Based Reconstruction of Extreme Cold-Season Precipitation and Number of

- Atmospheric River Landfalls Across the Southwestern United States. *Geophys. Res. Lett.* **45**, 5672-5680, doi:10.1029/2018gl078089 (2018).
- 30 Blöschl, G. *et al.* Changing climate shifts timing of European floods. *Science* **357**, 588-590, doi:10.1126/science.aan2506 (2017).
- 31 Meko, D. M., Stockton, C. W. & Boggess, W. R. The Tree-Ring Record of Severe Sustained Drought. *JAWRA Journal of the American Water Resources Association* **31**, 789-801, doi:doi:10.1111/j.1752-1688.1995.tb03401.x (1995).
- 32 Boé, J., Terray, L., Habets, F. & Martin, E. Statistical and dynamical downscaling of the Seine basin climate for hydro-meteorological studies. *Int. J. Climatol.* **27**, 1643-1655, doi:10.1002/joc.1602 (2007).
- 33 Cook, E. R., Krusic, P. J. & Jones, P. D. Dendroclimatic signals in long tree-ring chronologies from the Himalayas of Nepal. *Int. J. Climatol.* **23**, 707-732, doi:10.1002/joc.911 (2003).
- 34 Liu, J., Yang, B., Huang, K. & Sonechkin, D. M. Annual regional precipitation variations from a 700 year tree-ring record in south Tibet, western China. *Clim. Res.* **53**, 25-41 (2012).
- 35 Li, J. *et al.* Moisture increase in response to high-altitude warming evidenced by tree-rings on the southeastern Tibetan Plateau. *Clim. Dyn.* **48**, 649-660, doi:10.1007/s00382-016-3101-z (2017).
- 36 Chowdhury, M. Q., De Ridder, M. & Beeckman, H. Climatic Signals in Tree Rings of *Heritiera fomes* Buch.-Ham. in the Sundarbans, Bangladesh. *PLOS ONE* **11**, e0149788, doi:10.1371/journal.pone.0149788 (2016).
- 37 Thapa, U., St. George, S., Kharal, D. & Gaire, N. Tree growth across the Nepal Himalaya during the last four centuries. *Progress in Physical Geography: Earth and Environment* **41**, 478-495, doi:10.1177/0309133317714247 (2017).
- 38 Islam, M., Rahman, M. & Bräuning, A. Growth-Ring Boundary Anatomy and Dendrochronological Potential in a Moist Tropical Forest in Northeastern Bangladesh. *Tree-Ring Research* **74**, 76-93, doi:10.3959/1536-1098-74.1.76 (2018).
- 39 Maxwell, J. T., Harley, G. L. & Rahman, A. F. Annual Growth Rings in Two Mangrove Species from the Sundarbans, Bangladesh Demonstrate Linkages to Sea-Level Rise and Broad-Scale Ocean-Atmosphere Variability. *Wetlands* **38**, 1159-1170, doi:10.1007/s13157-018-1079-5 (2018).
- 40 Shi, C. *et al.* The response of relative humidity to centennial-scale warming over the southeastern Tibetan Plateau inferred from tree-ring width chronologies. *Clim. Dyn.* **51**, 3735-3746, doi:10.1007/s00382-018-4107-5 (2018).
- 41 Knutti, R. & Sedláček, J. Robustness and uncertainties in the new CMIP5 climate model projections. *Nature Climate Change* **3**, 369-373, doi:10.1038/nclimate1716 (2013).
- 42 Immerzeel, W. Historical trends and future predictions of climate variability in the Brahmaputra basin. *Int. J. Climatol.* **28**, 243-254, doi:10.1002/joc.1528 (2008).
- 43 Mahanta, C. & Saikia, L. The Brahmaputra and other rivers of the North-east. *Living rivers, Dying Rivers*, 155 (2015).
- 44 Mondal, M. S., Rahman, M. A., Mukherjee, N., Huq, H. & Rahman, R. Hydro-climatic hazards for crops and cropping system in the chars of the Jamuna River and potential adaptation options. *Nat. Hazards* **76**, 1431-1455, doi:10.1007/s11069-014-1424-9 (2015).

- 45 Becker, M. *et al.* Water level changes, subsidence, and sea level rise in the Ganges–Brahmaputra–Meghna delta. *Proceedings of the National Academy of Sciences*, 201912921, doi:10.1073/pnas.1912921117 (2020).
- 46 Uhe, P. F. *et al.* Enhanced flood risk with 1.5 °C global warming in the Ganges–Brahmaputra–Meghna basin. *Environmental Research Letters* **14**, 074031, doi:10.1088/1748-9326/ab10ee (2019).
- 47 Immerzeel, W. W., Pellicciotti, F. & Bierkens, M. F. P. Rising river flows throughout the twenty-first century in two Himalayan glacierized watersheds. *Nat. Geosci.* **6**, 742, doi:10.1038/ngeo1896
<https://www.nature.com/articles/ngeo1896#supplementary-information> (2013).
- 48 Lutz, A. F., Immerzeel, W. W., Shrestha, A. B. & Bierkens, M. F. P. Consistent increase in High Asia's runoff due to increasing glacier melt and precipitation. *Nature Climate Change* **4**, 587, doi:10.1038/nclimate2237
<https://www.nature.com/articles/nclimate2237#supplementary-information> (2014).
- 49 Nepal, S. & Shrestha, A. B. Impact of climate change on the hydrological regime of the Indus, Ganges and Brahmaputra river basins: a review of the literature. *International Journal of Water Resources Development* **31**, 201-218, doi:10.1080/07900627.2015.1030494 (2015).
- 50 Kale, V. S., Ely, L. L., Enzel, Y. & Baker, V. R. in *Geomorphology and Natural Hazards* (ed M. Morisawa) 157-168 (Elsevier, 1994).
- 51 Ely, L. L., Enzel, Y., Baker, V. R., Kale, V. S. & Mishra, S. Changes in the magnitude and frequency of late Holocene monsoon floods on the Narmada River, central India. *GSA Bulletin* **108**, 1134-1148, doi:10.1130/0016-7606(1996)108<1134:Citmaf>2.3.Co;2 (1996).
- 52 Kale, V. S., Ely, L. L., Enzel, Y. & Baker, V. R. Palaeo and historical flood hydrology, Indian Peninsula. *Geological Society, London, Special Publications* **115**, 155-163, doi:10.1144/gsl.Sp.1996.115.01.12 (1996).
- 53 Thomas, P. J., Juyal, N., Kale, V. S. & Singhvi, A. K. Luminescence chronology of late Holocene extreme hydrological events in the upper Penner River basin, South India. *J. Quat. Sci.* **22**, 747-753, doi:10.1002/jqs.1097 (2007).
- 54 Wilhelm, B. *et al.* Interpreting historical, botanical, and geological evidence to aid preparations for future floods. *WIREs Water* **6**, e1318, doi:10.1002/wat2.1318 (2019).
- 55 Speer, J. H. *et al.* Flood history and river flow variability recorded in tree rings on the Dhur River, Bhutan. *Dendrochronologia* **56**, 125605, doi:<https://doi.org/10.1016/j.dendro.2019.125605> (2019).
- 56 Ballesteros-Cánovas, J. A., Stoffel, M., St George, S. & Hirschboeck, K. A review of flood records from tree rings. *Progress in Physical Geography: Earth and Environment* **39**, 794-816, doi:10.1177/0309133315608758 (2015).
- 57 Allen, K. J., Hope, P., Lam, D., Brown, J. R. & Wasson, R. J. Improving Australia's flood record for planning purposes – can we do better? *Australasian Journal of Water Resources*, 1-10, doi:10.1080/13241583.2020.1745735 (2020).
- 58 Cook, E. R. & Kairiukstis, L. *Methods of Dendrochronology: Applications in the Environmental Sciences*. (Springer Science & Business Media, 1990).
- 59 Fritts, H. Tree rings and climate, 567 pp. *Academic, San Diego, Calif* (1976).

- 60 Melvin, T. M. & Briffa, K. R. A “signal-free” approach to dendroclimatic
standardisation. *Dendrochronologia* **26**, 71-86,
[doi:https://doi.org/10.1016/j.dendro.2007.12.001](https://doi.org/10.1016/j.dendro.2007.12.001) (2008).
- 61 Cook, E. R. *et al.* Five centuries of Upper Indus River flow from tree rings. *J.*
Hydrol. **486**, 365-375, [doi:https://doi.org/10.1016/j.jhydrol.2013.02.004](https://doi.org/10.1016/j.jhydrol.2013.02.004) (2013).
- 62 Wigley, T. M., Briffa, K. R. & Jones, P. D. On the average value of correlated time
series, with applications in dendroclimatology and hydrometeorology. *J. Glim.*
Appl. Meteorol. **23**, 201-213 (1984).
- 63 Efron, B. & Tibshirani, R. Bootstrap Methods for Standard Errors, Confidence Intervals,
and Other Measures of Statistical Accuracy. *Statistical Science* **1**, 54-75 (1986).

Reviewer #1 (Remarks to the Author):

NCOMMS-20-04754A "Seven centuries of reconstructed Brahmaputra River discharge demonstrate underestimated high discharge and flood hazard frequency"

Submitted to Nature Communications by Rao et al.

Although the authors have provided a substantial revision of their manuscript and detailed replies to my comments and questions, I'm still not convinced by their interpretation of the results. My concern is with regards to following points:

i) The authors recognize the uncertainty in the underlying tree-ring streamflow estimates, which is inherent in all reconstruction methods and I aware with that. However, their goal is not only to provide valuable information about the past discharge variability but also to perform a comparative analysis of annual reconstructions of the past extreme flood events. I do not find, from the authors' reply, an argument that would be persuasive enough to admit that 53% of variance explained by the tree-ring reconstruction and 24% of the shared variance in PC1 (climate information), are sufficient to establish a causal relationship about intensity of the extreme flood periods. The contribution of the process(es) explaining other 47% of the variance and not taken into account in the reconstruction should be evaluated. Otherwise, the authors' assertions suffer from the lack of credibility.

ii) Moreover, the authors draw an important conclusion that the mean discharge over 1309-2004 is significantly higher than that during 1956-2011. Their conclusion is supported by the t-test statistics. However, as I've pointed previously, the authors applied the t-test without discussing properly the basic assumptions for the t-statistics to be valid in this study. A difference in means cannot be admitted without careful inspection of the statistical features of both (instrumental and reconstructed) records.

Finally, I'd like to encourage the authors: this is a very interesting article, but its current version did not convince me that a causal relationship exists. As I wrote in my first review, these points were among the 2 crucial points for the manuscript to be accepted in Nature Communications.

Reviewer #2 (Remarks to the Author):

I have read the authors' responses to my (and the other reviewers') comments, as well as the revised manuscript. I appreciate the thorough and thoughtful additional analyses and clarifications. Clearly this is a complex system and I believe that the authors have a done a reasonable job of discussing the shortcomings of the research. No doubt more sophisticated analyses will be possible in the future, but for now this work is an excellent start. Given the practical relevance of the findings I think it is important to publish this work. I think the manuscript is in more or less in publishable form.

Sincerely,
Matthew Therrell
University of Alabama

Reviewer #3 (Remarks to the Author):

Authors have done a great work in this review process.

Previously, I had concerns related to (i) the improper use (and focus) of the term “flood risk” in the ms, (ii) the assumed robustness of conclusion with a reconstruction of 50% of variance; and (iii) the definition of flood event based on tree-growths. I have realized that some of these concerns were also common among the other reviewers.

In the current version, I really recognize the great effort done by the authors to address these and other suggestions. In particular, it was important for me to use properly the concept of “risk” and highlights the limitation of the methods, even more given the huge impact of floods in this region of the world. I am satisfied with the newer text that the focus on “hazard” and more explicitly the recognition of tree-rings “as proxy of hazard”. I also appreciate a clearer text, specifically regarding the definition of flood events. I also recognize that the authors answered properly other reviewers, so not more issues from my side.

Please find below our point-by-point response (in blue) to the comments by Reviewer #1 (in red). In our revision we also include a ‘track changes’ version of our updated manuscript.

Reviewer #1 (Remarks to the Author):

NCOMMS-20-04754A "Seven centuries of reconstructed Brahmaputra River discharge demonstrate underestimated high discharge and flood hazard frequency" Submitted to Nature Communications by Rao et al. Although the authors have provided a substantial revision of their manuscript and detailed replies to my comments and questions, I'm still not convinced by their interpretation of the results. My concern is with regards to following points:

i) The authors recognize the uncertainty in the underlying tree-ring streamflow estimates, which is inherent in all reconstruction methods and I aware with that. However, their goal is not only to provide valuable information about the past discharge variability but also to perform a comparative analysis of annual reconstructions of the past extreme flood events. I do not find, from the authors' reply, an argument that would be persuasive enough to admit that 53% of variance explained by the tree-ring reconstruction and 24% of the shared variance in PC1 (climate information), are sufficient to establish a causal relationship about intensity of the extreme flood periods. The contribution of the process(es) explaining other 47% of the variance and not taken into account in the reconstruction should be evaluated. Otherwise, the authors' assertions suffer from the lack of credibility.

Thank you for your helpful feedback and comments. We respond to this concern in two parts. In the first section of our response, we discuss how information contained in our reconstruction of past mean July-August-September (JAS) Brahmaputra River discharge can be used to assess the increased likelihood of past flood hazard. In the second section, we evaluate the climate variables that drive mean JAS instrumental discharge and assess how reliably the reconstruction captures these relationships.

We would also like to note that the calibration R^2 of our updated reconstruction based on the previous revision is 65.6%. As we highlighted in our previous point-by-point response, this R^2 is well situated in the upper range of accepted skill for tree-ring hydroclimate reconstructions across the world, and more specifically in Monsoon Asia¹⁻¹⁰. Further, in our tree-ring model we actually used the first 9 principal component timeseries (PCs) of the 28 tree-ring series that have eigenvalues > 1 as potential predictors of streamflow. These 9 PCs cumulatively explain 83.6% of shared variance of the 28 predictors. We mainly discussed PC1 and PC2 in the manuscript as they explained the highest percentage of the shared variance (24.9% and 13.6%, respectively). For transparency in our methods, our full reconstruction R code is provided along with the manuscript.

We have clarified in our previous submitted manuscript revisions that we are not reconstructing all flood events, flood risk, or performing a comparative analysis of annual reconstructions of the past extreme flood events. ***Rather, we are reconstructing monsoon season July-September (JAS) discharge, and using this information to show how the likelihood of flood hazards has changed over time.*** We believe this is an appropriate use of our reconstruction based on the analysis outlined in Fig. 1 below (same as Fig. 2b from the main text). Of the 55 years of instrumental data between 1956-2011 (1971 missing) there are a total of 13 years when discharge was greater than or equal

to 48,800 m³/s (Fig. 1, see Methods for explanation about how this threshold was determined). Of these 13 years, 6 were documented floods. Therefore, in the instrumental data we can consider the $P(\text{flood}|Q \geq 48,800 \text{ m}^3/\text{s}) = 6/13$. However, considering all 55 years of data, the probability of a flood event is $P(\text{flood}) = 6/55$. The odds ratio^{11,12} of a flood event given that discharge $\geq 48,800 \text{ m}^3/\text{s}$ in the instrumental data can therefore be calculated as $P(\text{flood}|Q \geq 48,800 \text{ m}^3/\text{s})/P(\text{flood}) = (6/13)/(6/55) = 4.23$. This odds ratio is much higher than 1 (i.e. by pure chance alone). This suggests that information contained in our reconstruction of past JAS discharge, and in particular a wetter mean state condition since 1309 C.E., does contain information that can be useful to interpret the likelihood of flood hazards.

Fig. 1 (same as main text Fig. 2) A scatter plot of mean JAS discharge against maximum 10-day mean daily discharge. The six flood years are highlighted in red. The bootstrapped Pearson and Spearman rank correlations are calculated as the median and 5th and 95th percentile of 1,000 draws with replacement. The grey uncertainty envelope ($\pm 2\sigma$) is derived from the best-fit linear regression (blue line). The vertical dashed line highlights mean JAS discharge in 2007.

Further, we show that high flow years in the instrumental data are associated with flood hazards (Fig. 1 above), and use this relationship to place recent decades of streamflow and flood hazards in a longer-term context based on the return interval analyses in Fig. 4 of the main text. Thus, the implications of our study are primarily regarding the likelihood of present-day flood hazard in light of the wetter long-term context indicated by our reconstruction. We certainly agree with the reviewer that it would be very informative to reconstruct past flood events, for example by using documentary sources¹³, tree-ring flood scars¹⁴, and other paleo-proxies. We highlight this in our manuscript (see manuscript text below). We also do not argue that the occurrence of a year with discharge greater than 48,800 m³/s in our reconstruction guarantees a flood event. Indeed, that is not the case even in the instrumental record as Fig. 1 indicates. So, $Q \geq 48,800 \text{ m}^3/\text{s}$ discharge is not always a flood event, but the probability of it being a flood event meaningfully increases as the odds ratio indicates. As we note in the updated manuscript, and this was the main concern raised by Review #3, the occurrence of a ‘flood event’ itself is closely tied to exposure and vulnerability. We have modified the language used in our manuscript accordingly.

Lines 285-296 - “While our recurrence interval analysis focusses on the frequency of high discharge that is associated with the likelihood of flood hazard, many other factors play a role in determining whether high discharge translates to a flood event. These may include rainfall intensity and pattern, landscape heterogeneities, antecedent soil moisture conditions, and land use and forest cover change⁶¹⁻⁶³. Our return interval analyses also rely on the assumption that these high discharges will continue to be associated with an increased likelihood of flood hazard in the future, disregarding (for example) potential changes in policy, land use, or infrastructure that may ameliorate ‘flood risk’. The occurrence of a flood event that impacts society is however closely intertwined with highly localised human exposure and vulnerability^{64,65}. Therefore, our calculations of underestimated high discharge and associated likelihood of flood hazard in the return interval analyses in Fig. 3b only contributes one component of the multiple dimensions of flood risk.”

Turning now to the second part of our response - to evaluate in further detail the climatic processes that can influence mean JAS instrumental and reconstructed discharge, we computed monthly correlations between streamflow and upper basin climate variables (Fig. 2 below). The climate variables we explored include a spatial average of: i. monthly simulated runoff, ii. precipitation, iii. 2m air temperature, and iv. snow depth equivalent from the ERA-5¹⁵ dataset (Figure 2a and 2b). We also used other climate variables including, i. the self-calibrating Palmer Drought Severity Index (scPDSI¹⁶), ii. precipitation, iii. temperature, and iv. potential evapotranspiration (PET) from the CRU¹⁷ dataset. The spatial area used for calculating the climate averages was 88.5-96.5°E and 27-30.5°N. This is the same region for which ERA-5 simulated runoff was averaged in Supplementary Fig. S3 (grey box), to compare simulated upper basin runoff against observations of discharge at Bahadurabad, Bangladesh.

We found that both the instrumental observations and the reconstruction of JAS discharge correlated significantly and positively with upper basin precipitation in the months of June-August (JJA) and July-September (JAS) for the ERA-5 and CRU climate datasets ($p < 0.01$). However, this relationship was weaker for reconstructed discharge than for instrumental discharge (right vs left panel, Fig. 2). This suggests that while the reconstruction does capture the upper basin climate relationship inherent in the instrumental data, it does so imperfectly. This is a likely source of ‘unexplained variance’.

Using the ERA-5 dataset, mean JAS discharge at Bahadurabad is also correlated positively to upper basin snow depth equivalent in January and February ($p < 0.1$), inversely to January temperature ($p < 0.05$), positively to May precipitation ($p < 0.1$) and May temperature ($p < 0.05$), and to June snow depth equivalent ($p < 0.1$). In addition to these relationships, mean JAS discharge at Bahadurabad is significantly inversely correlated to upper basin PET in the CRU dataset. This suggests that increased PET in the upper basin can decrease discharge due to increased losses from evapotranspiration leading to a lower runoff efficiency. Taken together, the relationships with upper basin climate data (from ERA-5 and CRU) suggest that JAS streamflow is also influenced by climate in earlier seasons, including January-February winter snowpack accumulation, pre-monsoon precipitation in May, summer snowpack melt, and summer evapotranspiration. However, all of these climate-streamflow relationships with the instrumental data are weaker in our reconstruction. This could contribute to additional ‘unexplained variance’.

Another potential bias in our analyses could derive from our focus on upper basin processes in driving lower basin discharge. A majority of the tree-ring predictors that we use in our reconstruction are also situated in the upper basin. While larger spatial scale upper basin processes are the main drivers of Brahmaputra River discharge at Bahadurabad (Fig 2a, JAS discharge vs JAS ERA-5 Runoff), it is possible that our reconstruction does not capture lower basin processes that influence streamflow very well.

Fig. 2 (now also Fig. S4) Correlation response function plot for mean July-September (JAS) instrumental and reconstructed discharge of the Brahmaputra River at Bahadurabad (Bangladesh) against upper basin (88.5-96.5°E and 27-30.5°N) monthly climate variables from the ERA-5 (parts **a-b**) CRU (parts **c-d**) datasets. The climate variables used from the ERA-5 dataset (parts **a-b**) include monthly simulated runoff, precipitation, 2m air temperature, and snow depth equivalent averaged over the upper basin. The climate variables from the CRU dataset include scPDSI, precipitation, temperature, and potential evapotranspiration (PET) also averaged over the upper basin. The left panels (a and c) are for instrumental observations between 1981-2011 and 1956-2011 respectively. The right panels (b and d) are for reconstructed discharge between 1981-2004 and 1956-2004 respectively. The six horizontal dashed lines in each plot indicate three different thresholds for each monthly correlation to be statistically significant using a 2-tailed t-test. The median correlation and the error bars around each correlation bar are computed from 1,000 bootstrapped draws with replacement from the observed/reconstructed discharge series and the climate series. The last two columns of each subplot are the correlation between mean JAS observed/reconstructed discharge and mean upper basin climate averaged between July-August (JJA) and JAS. scPDSI - self calibrating Palmer Drought Severity Index¹⁶.

We now include the above analyses along with this explanation in Supplementary Fig. S4 and reference the figure and in the main manuscript.

Updated lines 143-147: “A comparison of the correlations between reconstructed discharge and upper basin climate variables indicates that the reconstruction captures the climate-streamflow relationships inherent in the instrumental observations (Supplementary Fig. S4). However, these climate-streamflow relationships are slightly weaker for reconstructed discharge than for instrumental discharge.”

ii) Moreover, the authors draw an important conclusion that the mean discharge over 1309-2004 is significantly higher than that during 1956-2011. Their conclusion is supported by the t-test statistics. However, as I've pointed previously, the authors applied the t-test without discussing properly the basic assumptions for the t-statistics to be valid in this study. A difference in means cannot be admitted without careful inspection of the statistical features of both (instrumental and reconstructed) records.

We have modified our analyses to use a non-parametric block bootstrap instead of a t-test. In our analysis, we now compare the instrumental mean for 7 different time intervals (1956-1986, 1956-1998, 1956-2004, 1956-2011, 1987-1998, 1987-2004, and 1987-2011) against a distribution of the mean of 10,000 randomly sampled blocks from the reconstruction (Fig. 3). The block length chosen in each resampling was consistent with the number of years of instrumental data used for the comparison. We found that the instrumental mean between 1956-1986, 1956-1998, 1956-2004, and 1956-2011 were consistently drier than the 5th percentile of the mean of 10,000 random draws of blocks of 31, 43, 49, and 56 years from the reconstruction. However, we also emphasise that this result is primarily driven by the extreme dry conditions in the first 31 of years of the instrumental data between 1956-1986. We demonstrate this by comparing the instrumental mean between 1987-1998, 1987-2004, and 1987-2011 to the mean of 10,000 random draws of blocks of 12, 28, and 25 years from the reconstruction. We note that these results are consistent the results we reported in the manuscript using t-tests.

Updated lines 168-171: “We also compared mean JAS discharge during these 4 intervals (i.e. 1956-1986, 1956-1998, 1956-2004, and 1956-2011) to random block bootstrap draws from the full reconstruction. We found all four intervals to be significantly drier ($p < 0.05$) than the reconstruction (Supplementary Fig. S7).”

In addition to including Fig. 3 as Supplementary Fig. S7 in the manuscript, we have modified the text in the manuscript as follows to highlight how this result is influenced by the dry period between 1956-1986 C.E.:

Updated lines 181-188: “We do note that both instrumental observations and the reconstructions show a return to wetter conditions starting in 1987 (also see refs. ^{7,18}). For example, more recent instrumental observations of discharge between 1987-1998 C.E., 1987-2004 C.E., and 1987-2011 C.E. are relatively wetter than the instrumental data prior to 1987 and fall in the 59th, 48th, and 39th percentile of full reconstruction (Supplementary Fig. S7).”

Fig. 3 (Now also Fig S7) Comparison between mean instrumental JAS Brahmaputra River discharge at 7 different time intervals (1956-1986; 1956-1998; 1956-2004; 1956-2011; 1987-1998; 1987-2004; and 1987-2011) as filled blue dots against distributions of the mean reconstructed discharge in 10,000 random draws of blocks of same length from the reconstruction. The block length used in each draw is mentioned below each box plot. The two red horizontal lines indicate the threshold for mean discharge to be significantly drier than reconstructed discharge at $p < 0.05$ and $p < 0.01$. The plot suggests that the first 31 years of instrumental discharge between 1956-1986 were exceptionally dry ($p < 0.05$) while discharge since 1987 aligns more closely with mean reconstructed discharge rates in the context of the past 7 centuries.

Finally, I'd like to encourage the authors: this is a very interesting article, but its current version did not convince me that a causal relationship exists. As I wrote in my first review, these points were among the 2 crucial points for the manuscript to be accepted in Nature Communications.

We assume here that the reviewer is not referring to the physical basis for the applicability of tree rings to reconstruct past streamflow (and climate). Schulman 1942¹⁹, Stockton 1971²⁰, Fritts 1976²¹, Stockton and Jacoby 1976²², and Meko 1995²³ are some early references for the investigation of tree-ring and streamflow/climate relationships. <https://www.treeflow.info> and <http://paleoflow.org/> include many other examples of successful reconstructions of streamflow by tree-rings. The references we cite in the paper also show that such streamflow reconstructions can be successfully done in the mountains of High Asia^{1,6,7,24,25}. There is also a well established relationship between total discharge and flood events for large river basins in the world²⁶, including the Brahmaputra River specifically²⁷⁻²⁹. In our work, we connect these two well established relationships to investigate whether we might currently be underestimating the likelihood of flood hazard as the instrumental period is exceptionally dry in the context of the past seven centuries.

.....
 Reviewer #2 (Remarks to the Author):

I have read the authors' responses to my (and the other reviewers') comments, as well as the revised manuscript. I appreciate the thorough and thoughtful additional analyses and clarifications. Clearly this is a complex system and I believe that the authors have done a reasonable job of discussing the shortcomings of the research. No doubt more sophisticated analyses will be possible in the future, but for now this work is an excellent start. Given the practical relevance of the findings I

think it is important to publish this work. I think the manuscript is in more or less in publishable form.

Sincerely,
Matthew Therrell
University of Alabama

Reviewer #3 (Remarks to the Author):

Authors have done a great work in this review process.

Previously, I had concerns related to (i) the improper use (and focus) of the term “flood risk” in the ms, (ii) the assumed robustness of conclusion with a reconstruction of 50% of variance; and (iii) the definition of flood event based on tree-growths. I have realized that some of these concerns were also common among the other reviewers.

In the current version, I really recognize the great effort done by the authors to address these and other suggestions. In particular, it was important for me to use properly the concept of “risk” and highlights the limitation of the methods, even more given the huge impact of floods in this region of the world. I am satisfied with the newer text that the focus on “hazard” and more explicitly the recognition of treerings “as proxy of hazard”. I also appreciate a clearer text, specifically regarding the definition of flood events. I also recognize that the authors answered properly other reviewers, so not more issues from my side.

We thank reviewers #2 and #3 for helping us considerably improve our manuscript with their thoughtful feedback and comments.

References

- 1 Gou, X. *et al.* Streamflow variations of the Yellow River over the past 593 years in western China reconstructed from tree rings. *Water Resour. Res.* **43**, doi:10.1029/2006wr005705 (2007).
- 2 Buckley, B. M. *et al.* Climate as a contributing factor in the demise of Angkor, Cambodia. *Proceedings of the National Academy of Sciences* **107**, 6748-6752, doi:10.1073/pnas.0910827107 (2010).
- 3 Feng, S., Hu, Q., Wu, Q. & Mann, M. E. A Gridded Reconstruction of Warm Season Precipitation for Asia Spanning the Past Half Millennium. *J. Clim.* **26**, 2192-2204, doi:10.1175/jcli-d-12-00099.1 (2013).
- 4 Yang, B. *et al.* A 3,500-year tree-ring record of annual precipitation on the northeastern Tibetan Plateau. *Proceedings of the National Academy of Sciences* **111**, 2903-2908, doi:10.1073/pnas.1319238111 (2014).
- 5 Hessl, A. E. *et al.* Past and future drought in Mongolia. *Science Advances* **4**, e1701832, doi:10.1126/sciadv.1701832 (2018).
- 6 Li, J. *et al.* Deciphering Human Contributions to Yellow River Flow Reductions and Downstream Drying Using Centuries-Long Tree Ring Records. *Geophys. Res. Lett.* **46**, 898-905, doi:10.1029/2018gl081090 (2019).
- 7 Chen, Y. *et al.* Precipitation variations recorded in tree rings from the upper Salween and Brahmaputra River valleys, China. *Ecol. Indicators* **113**, 106189, doi:<https://doi.org/10.1016/j.ecolind.2020.106189> (2020).

- 8 He, M., Bräuning, A., Grießinger, J., Hochreuther, P. & Wernicke, J. May–June drought reconstruction over the past 821 years on the south-central Tibetan Plateau derived from tree-ring width series. *Dendrochronologia* **47**, 48-57, doi:<https://doi.org/10.1016/j.dendro.2017.12.006> (2018).
- 9 Wang, J., Yang, B. & Ljungqvist, F. C. Moisture and Temperature Covariability over the Southeastern Tibetan Plateau during the Past Nine Centuries. *J. Clim.* **0**, null, doi:10.1175/jcli-d-19-0363.1 (2020).
- 10 Ljungqvist, F. C. *et al.* Ranking of tree-ring based hydroclimate reconstructions of the past millennium. *Quat. Sci. Rev.* **230**, 106074, doi:<https://doi.org/10.1016/j.quascirev.2019.106074> (2020).
- 11 Haer, T., Botzen, W. J. W. & Aerts, J. C. J. H. The effectiveness of flood risk communication strategies and the influence of social networks—Insights from an agent-based model. *Environmental Science & Policy* **60**, 44-52, doi:<https://doi.org/10.1016/j.envsci.2016.03.006> (2016).
- 12 Bubeck, P., Botzen, W. J. W., Kreibich, H. & Aerts, J. C. J. H. Detailed insights into the influence of flood-coping appraisals on mitigation behaviour. *Global Environmental Change* **23**, 1327-1338, doi:<https://doi.org/10.1016/j.gloenvcha.2013.05.009> (2013).
- 13 Blöschl, G. *et al.* Current European flood-rich period exceptional compared with past 500 years. *Nature* **583**, 560-566, doi:10.1038/s41586-020-2478-3 (2020).
- 14 Ballesteros-Cánovas, J. A., Stoffel, M., St George, S. & Hirschboeck, K. A review of flood records from tree rings. *Progress in Physical Geography: Earth and Environment* **39**, 794-816, doi:10.1177/0309133315608758 (2015).
- 15 Copernicus Climate Change Service (C3S). C3S ERA5-Land reanalysis . Copernicus Climate Change Service <https://cds.climate.copernicus.eu/cdsapp#!/home>. (2019).
- 16 van der Schrier, G., Barichivich, J., Briffa, K. R. & Jones, P. D. A scPDSI-based global data set of dry and wet spells for 1901–2009. *Journal of Geophysical Research: Atmospheres* **118**, 4025-4048, doi:10.1002/jgrd.50355 (2013).
- 17 Harris, I., Jones, P. D., Osborn, T. J. & Lister, D. H. Updated high-resolution grids of monthly climatic observations – the CRU TS3.10 Dataset. *Int. J. Climatol.* **34**, 623-642, doi:10.1002/joc.3711 (2014).
- 18 Shi, C. *et al.* The response of relative humidity to centennial-scale warming over the southeastern Tibetan Plateau inferred from tree-ring width chronologies. *Clim. Dyn.* **51**, 3735-3746, doi:10.1007/s00382-018-4107-5 (2018).
- 19 Schulman, E. A Tree-Ring History of Runoff of the Colorado River, 1366-1941. *Report, Bureau of Power and Light, Los Angeles, California* (1942).
- 20 Stockton, C. W. (The University of Arizona., 1971).
- 21 Fritts, H. Tree rings and climate, 567 pp. *Academic, San Diego, Calif* (1976).
- 22 Stockton, C. W. & Jacoby, G. C. Long-term surface water supply and streamflow levels in the upper Colorado River basin. Lake Powell Research Project. *Bulletin* **18**, 70 (1976).
- 23 Meko, D. M., Stockton, C. W. & Boggess, W. R. The Tree-Ring Record of Severe Sustained Drought. *JAWRA Journal of the American Water Resources Association* **31**, 789-801, doi:10.1111/j.1752-1688.1995.tb03401.x (1995).
- 24 Cook, E. R. *et al.* Five centuries of Upper Indus River flow from tree rings. *J. Hydrol.* **486**, 365-375, doi:<https://doi.org/10.1016/j.jhydrol.2013.02.004> (2013).
- 25 Rao, M. P. *et al.* Six Centuries of Upper Indus Basin Streamflow Variability and Its Climatic Drivers. *Water Resour. Res.* **54**, 5687-5701, doi:10.1029/2018wr023080 (2018).

- 26 Milly, P. C. D., Wetherald, R. T., Dunne, K. A. & Delworth, T. L. Increasing risk of great floods in a changing climate. *Nature* **415**, 514-517, doi:10.1038/415514a (2002).
- 27 Immerzeel, W. Historical trends and future predictions of climate variability in the Brahmaputra basin. *Int. J. Climatol.* **28**, 243-254, doi:10.1002/joc.1528 (2008).
- 28 Jian, J., Webster, P. J. & Hoyos, C. D. Large-scale controls on Ganges and Brahmaputra river discharge on intraseasonal and seasonal time-scales. *Q. J. Roy. Meteorol. Soc.* **135**, 353-370, doi:10.1002/qj.384 (2009).
- 29 Webster, P. J. *et al.* Extended-Range Probabilistic Forecasts of Ganges and Brahmaputra Floods in Bangladesh. *Bull. Am. Meteorol. Soc.* **91**, 1493-1514, doi:10.1175/2010bams2911.1 (2010).

Reviewer #2 (Remarks to the Author):

At your request I have reviewed the authors' responses to the reviews of the manuscript (NCOMMS-20-04754-T) "Six centuries of reconstructed Brahmaputra River flow demonstrate underestimated flood risk" by Rao et al. As you note the authors' responses to the suggestions offered by Reviewer #3 and myself were generally acceptable and I felt that the authors took our suggestions seriously and made substantial improvements to the manuscript. My reading of the comments of Reviewer #1 on the revision of the manuscript suggests that the primary issue R1 has with the research is the strength (or rather perceived lack) of the statistical relationship as well as an unconvincing mechanistic explanation for the same. I understand how one might have this perception given that the relationship between tree growth and flooding is not "direct" but is rather based on a chain of moderately complex biogeophysical processes.

However, as the authors note, these processes and the embedded relationships are fairly well understood and are demonstrably coherent not only in the study area but essentially across the world. Moreover the statistical strength of the relationship is very strong. Again, as the authors note, this is among the highest reported explanations of tree-ring reconstructed variance for streamflow that I can recall from the literature. I certainly understand and respect R1's perception, but I would suggest that the research presented in this manuscript is entirely in line with the published literature on the subject and disciplinary norms. The authors have done a very good job of responding to R1's suggestions and concerns and I think this manuscript should be published in its current form.

Please let me know if I can provide any other information or answer any questions.

Sincerely,

Matthew Therrell

Professor